# The anterior insular cortex unilaterally controls feeding in response to aversive visceral stimuli in mice

Yu Wu[1,3], Changwan Chen[1,3], Ming Chen[1,3], Kai Qian[1], Xinyou Lv[1], Haiting Wang[1], Lifei Jiang[1], Lina Yu[1], Min Zhuo[2] & Shuang Qiu [1]*

Reduced food intake is common to many pathological conditions, such as infection and toxin exposure. However, cortical circuits that mediate feeding responses to these threats are less investigated. The anterior insular cortex (aIC) is a core region that integrates interoceptive states and emotional awareness and consequently guides behavioral responses. Here, we demonstrate that the right-side aIC CamKII$^+$ (aIC$^{CamKII}$) neurons in mice are activated by aversive visceral signals. Hyperactivation of the right-side aIC$^{CamKII}$ neurons attenuates food consumption, while inhibition of these neurons increases feeding and reverses aversive stimuli-induced anorexia and weight loss. Similar manipulation at the left-side aIC does not cause significant behavioral changes. Furthermore, virus tracing reveals that aIC$^{CamKII}$ neurons project directly to the vGluT2$^+$ neurons in the lateral hypothalamus (LH), and the right-side aIC$^{CamKII}$-to-LH pathway mediates feeding suppression. Our studies uncover a circuit from the cortex to the hypothalamus that senses aversive visceral signals and controls feeding behavior.

---

[1] Center for Neuroscience and Department of Anesthesiology of Second Affiliated Hospital, NHC and CAMS Key Laboratory of Medical Neurobiology, Zhejiang University School of Medicine, 310058 Hangzhou, Zhejiang, China. [2] Department of Physiology, Faculty of Medicine, University of Toronto, 1 King's College Circle, Toronto, ON M5S 1A8, Canada. [3]These authors contributed equally: Yu Wu, Changwan Chen, Ming Chen. *email: qiushly@zju.edu.cn

Whether to eat and how much to eat in any one meal are critical decisions for the health of all metazoan organisms. Under normal conditions, food intake and energy expenditure are balanced by well-defined hypothalamic neural circuits that maintain stable body weight[1–5]. However, food intake can be attenuated in pathological conditions, such as food poisoning, inflammation, or chemotherapy treatment that may elicit visceral malaise, indicating that some emergency-response circuits hijack the energy homeostasis system. However, the neural circuits involved in mediating feeding responses under these non-homeostatic conditions are comparatively elusive.

Insular cortex, especially the anterior part of the insular cortex (aIC), is a critical region for processing salient stimuli and orchestrating appropriate behavioral responses. The aIC receives heavy sensory inputs from the thalamus and serves as the primary visceral and gustatory cortex[6–14]. Neuroimaging studies in humans have revealed that the aIC responds not only to food cues, but also to multiple noxious stimuli that trigger pain, anxiety, and disgust[15]. Studies in patients with eating disorders, such as bulimia nervosa[16,17] and anorexia nervosa[18–20], have shown altered insula activation. Animal studies confirmed the activation of the insular cortex in response to anorexigenic signals in vivo[21–24]. Moreover, direct insular stimulation in humans or rodents induces visceral sensations such as nausea[25,26], whereas damage to the insular cortex blunts lithium-induced malaise[21] and blocks the behavioral expression of a conditioned taste aversion[27–29]. Although the role of the insular cortex in perceiving multiple aversive signals has been extensively studied, the behavioral responses under these pathological conditions, especially feeding behaviors, upon insular activation are not well studied. By now, a direct role of the insular cortex in controlling food intake has not yet been demonstrated in patients with insular damage[30] or in rodents with insular lesion or inactivation[21,31].

In this study, we demonstrate that the CamKII-positive neurons in the caudal segment of the right-side, but not the left-side, aIC respond to aversive visceral stimuli and suppress feeding via projections to the vGluT2-positive neurons in the lateral hypothalamus (LH). Thus, we identify a role of the aIC that conveys aversive visceral information to the LH to regulate food consumption.

## Results

**The right-side aIC is activated by aversive visceral stimuli.** To investigate the involvement of the aIC in processing aversive signals, we monitored Fos expression after intraperitoneal (i.p.) injection of anorexigenic signals, such as lithium chloride (LiCl), which induces nausea and visceral malaise, lipopolysaccharide (LPS), which triggers a wide range of inflammatory and sickness responses, and *cis*-diaminodichloroplatinum (Cisplatin)[32], which is a chemotherapy medication used to treat cancers but induces anorexia and cancer anorexia-cachexia. Immunostaining for Fos revealed that injection (i.p.) of LiCl, LPS, or Cisplatin-induced substantial Fos expression bilaterally in several brain regions, including parabrachial nucleus (PBN), nucleus of solitary tract (NTS), basolateral amygdaloid nucleus (BLA), medial prefrontal cortex (mPFC), LH, the parvicellular part of the ventroposteromedial nucleus of the thalamus (VPMpc), and central amygdala (CeA) (Two-way ANOVA, PBN, $F_{2,15} = 0.3023$, $P = 0.0681$; NTS, $F_{2,15} = 0.1253$, $P = 0.8831$; BLA, $F_{2,15} = 0.562$, $P = 0.5816$; mPFC, $F_{2,15} = 0.1878$, $P = 0.8307$, LH, $F_{2,15} = 0.7986$, $P = 0.4682$; VPMpc, $F_{2,15} = 0.1315$, $P = 0.8778$; CeA, $F_{2,15} = 0.1166$, $P = 0.9493$, Supplementary Fig. 1a, b). Of interest, LiCl, LPS, and Cisplatin produced a robust induction of Fos in the right, but not in the left aIC (Two-tailed unpaired $t$ test, Saline, $t_{10} = 0.6184$, $P = 0.5501$; LiCl, $t_{22} = 4.027$, $P = 0.0006$; LPS, $t_{18} = 6.946$,

$P < 0.0001$; Cisplatin, $t_{20} = 2.886$, $P = 0.0091$, Fig. 1a, b). Further analyses revealed that LiCl-induced or LPS-induced Fos-positive (Fos$^+$) neurons were mainly localized within the caudal segment between 1.26 mm before Bregma (Bregma +1.26) and 0.02 mm after Bregma (Bregma −0.02) of the right aIC (Two-way ANOVA, Saline, $F_{4,25} = 1.961$, $P = 0.1315$; LiCl, $F_{4,25} = 33.42$, $P < 0.0001$; LPS, $F_{4,25} = 6.957$, $P = 0.00061$, Supplementary Fig. 2a–d). In contrast, injection (i.p.) of cholecysokinin (CCK), a potential satiation factor, or ghrelin, a hunger hormone, resulted in a much less Fos expression (Two-tailed unpaired $t$ test, Ghrelin, $t_{10} = 1.607$, $P = 0.1391$; CCK, $t_{10} = 0.4837$, $P = 0.639$, Fig. 1a, b) and these Fos$^+$ neurons were diffusely distributed within both sides of the aIC (Two-way ANOVA, Ghrelin, $F_{4,25} = 1.641$, $P = 0.1612$; CCK, $F_{4,25} = 0.6909$, $P = 0.6052$, Supplementary Fig. 2e, f). Additionally, LiCl-induced and LPS-induced Fos$^+$ neurons were exclusively located in the agranular part of the caudal segment of the right aIC, whereas CCK-induced Fos$^+$ neurons were mainly located in the granular part and Ghrelin-induced Fos$^+$ neurons were diffusely distributed in this segment (one-way ANOVA, Saline, $F_{3,20} = 0.6521$, $P = 0.5909$; LiCl, $F_{3,20} = 3.416$, $P = 0.0372$; LPS, $F_{3,20} = 0.3912$, $P = 0.0212$; Ghrelin, $F_{3,20} = 0.6696$, $P = 0.5806$; CCK, $F_{3,20} = 2.617$, $P = 0.0477$, Supplementary Fig. 2g, h).

Moreover, no obvious Fos expression was observed in the left or right aIC in ad libitum-feeding mice, in 24-h fasted mice or in refeeding mice (two-tailed unpaired $t$ test, home caged, $t_{10} = 0.4953$, $P = 0.6311$; fasted, $t_{10} = 0.9209$, $P = 0.3788$, re-fed, $t_{10} = 0.5575$, $P = 0.5895$, Fig. 1c, d). We also investigated the mice with electric shock, which exerts no direct effect on the gastrointestinal tract, and observed significant Fos staining in the BLA bilaterally, but not in the aIC (two-way ANOVA, aIC, $F_{1,20} = 0.8639$, $P = 0.3637$; BLA, $F_{1,20} = 0.0216$, $P = 0.8846$, Supplementary Fig. 3a, b). Additionally, the right aIC was significantly activated no matter whether LiCl was injected (i.p.) in the left side or right side of the mouse body, suggesting that the unilateral activation of the aIC is not determined by injection site (two-way ANOVA, left side (i.p.), $F_{1,10} = 12.93$, $P = 0.0049$; right side (i.p.), $F_{1,10} = 47.22$, $P < 0.0001$, Supplementary Fig. 3c). These results indicate that aIC, especially the caudal segment of the right aIC, responds to multiple aversive visceral stimuli.

Notably, double immunohistochemical staining revealed that almost 82.6% of Fos$^+$ neurons activated by LiCl in the right aIC expressed CamKII, while only 4.1% of Fos$^+$ neurons expressed GAD67 (Fig. 1e, f). To confirm this result, we injected a recombinant adeno-associated virus (AAV) that expressed *Camk2a* promoter-controlled mCherry (AAV9-CamKIIα-mCherry) into the caudal segment of the right aIC of the wild-type mice. Three weeks later, we observed that 87.8% of Fos$^+$ neurons activated by LiCl were colocalized with mCherry-positive cells in this site (Supplementary Fig. 3d). Together, these results indicate that most of the activated neurons are excitatory neurons expressing CamKII.

To determine the dynamics of the CamKII-positive neurons in the caudal segment of the right or left aIC (the right or left aIC$^{CamKII}$ neurons) in vivo during exposure to an aversive stimulus, we performed fiber photometry recording (Fig. 1g, h, Supplementary Fig. 4a). AAV containing GCaMP6f (AAV9-mCamKIIα-GCaMP6f) was injected unilaterally into the caudal segment of the right aIC or the left aIC of wild-type mice (Fig. 1g, h). Strikingly, i.p. injection of LiCl (One-way ANOVA, $F_{3,12} = 8.409$, $P = 0.0028$, Fig. 1i, Supplementary Fig. 4b, c) or Cisplatin (One-way ANOVA, $F_{3,12} = 71.38$, $P < 0.0001$, Fig. 1j, Supplementary Fig. 4d, e) evoked sustained increase in the activity of the right aIC$^{CamKII}$ neuron, but not in the left aIC$^{CamKII}$ neurons, confirming that aversive visceral stimuli enhance the excitatory neuronal activity in the right aIC.

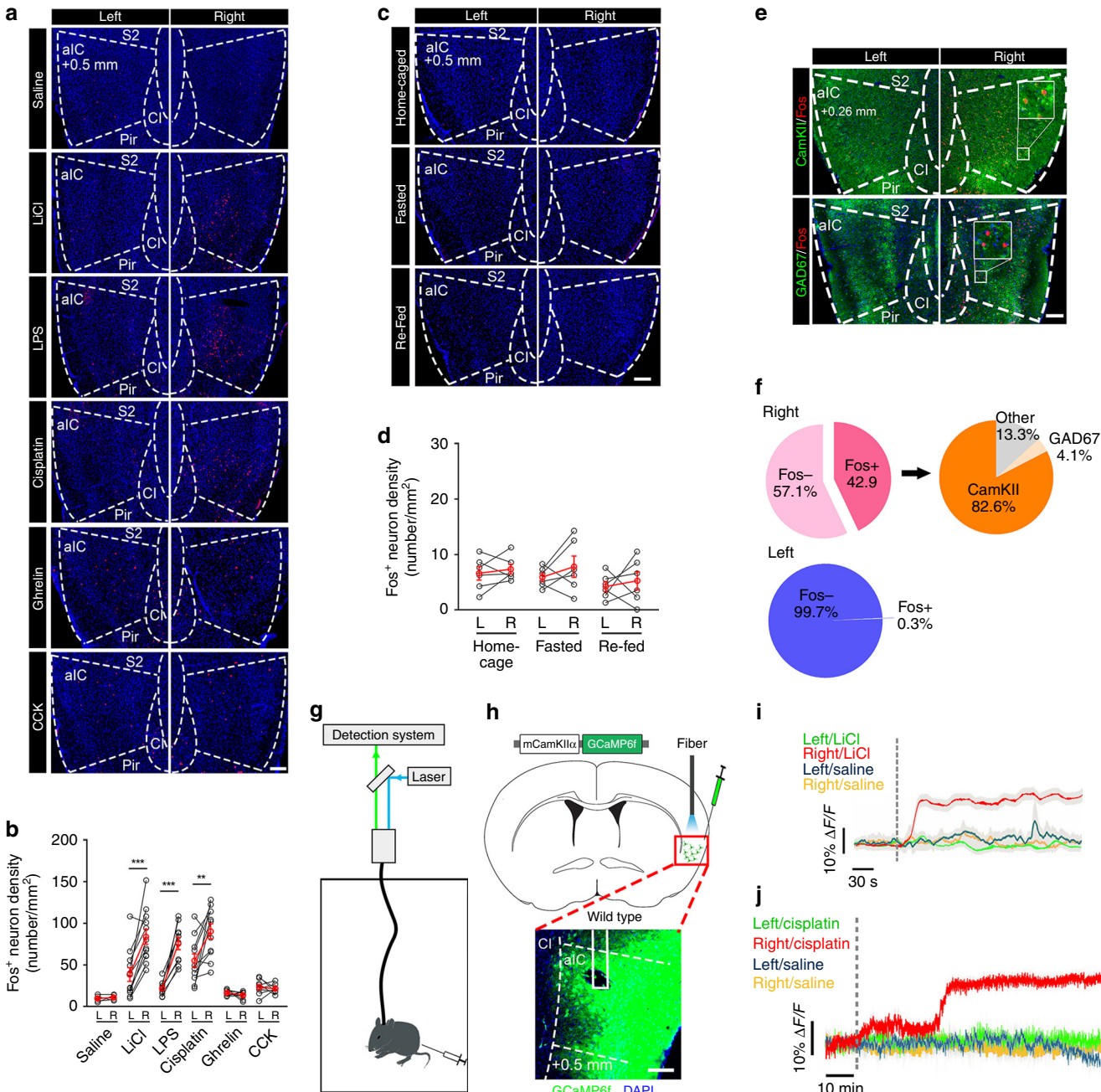

**Fig. 1 CamKII$^+$ neurons in the right aIC are activated by aversive visceral stimuli. a**, **b** Representative histology **a** and quantification **b** of Fos-like immunoreactivity in the left and the right aIC after intraperitoneal injection of saline ($n = 16$ mice), LiCl (150 mg/kg, $n = 21$ mice), cisplatin (4 mg/kg, $n = 11$ mice), LPS (0.1 mg/kg, $n = 14$ mice), Ghrelin (1 mg/kg, $n = 6$ mice), or CCK (5 µg/kg, $n = 6$ mice). Two-tailed unpaired $t$ test, scale bar, 250 µm. **c**, **d** Representative histology **c** and quantification **d** of Fos-like immunoreactivity in the left and the right aIC in fed or 24-h fasted mice with or without 3-h refeeding ($n = 6$ mice per group. Two-tailed unpaired $t$ test). Scale bar, 250 µm. **e**, **f** Co-expression **e** of CamKII or GAD67 (green) with Fos (red) in the right or left aIC injection with LiCl and quantification **f**. $n = 6$ mice per group. Scale bar, 100 µm. **g** Schema of the fiber photometry approach. **h** Image of GCaMP6f expression in the right aIC$^{CamKII}$, with the fiber track above. Scale bar, 500 µm. **i**, **j** Representative mean GCaMP6f fluorescence responses of both sides of the aIC$^{CamKII}$ neurons by injection of Saline, LiCl, or Cisplatin. i.p. intraperitoneal. $n = 4$ mice per group. For **b** and **d**, red line represents averaged data. **P < 0.01; ***P < 0.005. Data are presented as means ± SEM. Source data are provided as a Source Data file.

**Activating the right-side aIC$^{CamKII}$ neurons impairs feeding.** To explore whether the CamKII$^+$ neurons in the caudal segment of the right aIC mediate feeding behavior, we injected Cre-dependent AAV containing a double floxed inverted orientation (DIO) excitatory optogenetic receptor construct (AAV8-DIO-hChR2(H134R)-eGFP) into the caudal segment of the right aIC of *Camk2a-Cre* transgenic mice (Fig. 2a, Supplementary Fig. 5a). Whole-cell patch-clamp recordings in the acute brain slices

showed that action potentials of ChR2-expressing aIC$^{CamKII}$ neurons were triggered by 473-nm light pulses (Fig. 2b). We next performed a time-resolved analysis of feeding in mice that were deprived of food for 24 h (Fig. 2c). Photoactivation of the right aIC$^{CamKII}$ neurons at 20 Hz significantly decreased the amount of food intake (two-tailed unpaired $t$ test, $t_{17} = 2.385$, $P = 0.029$) and the percentage of time spent feeding (feeding time was determined as the time the mice successfully accomplished

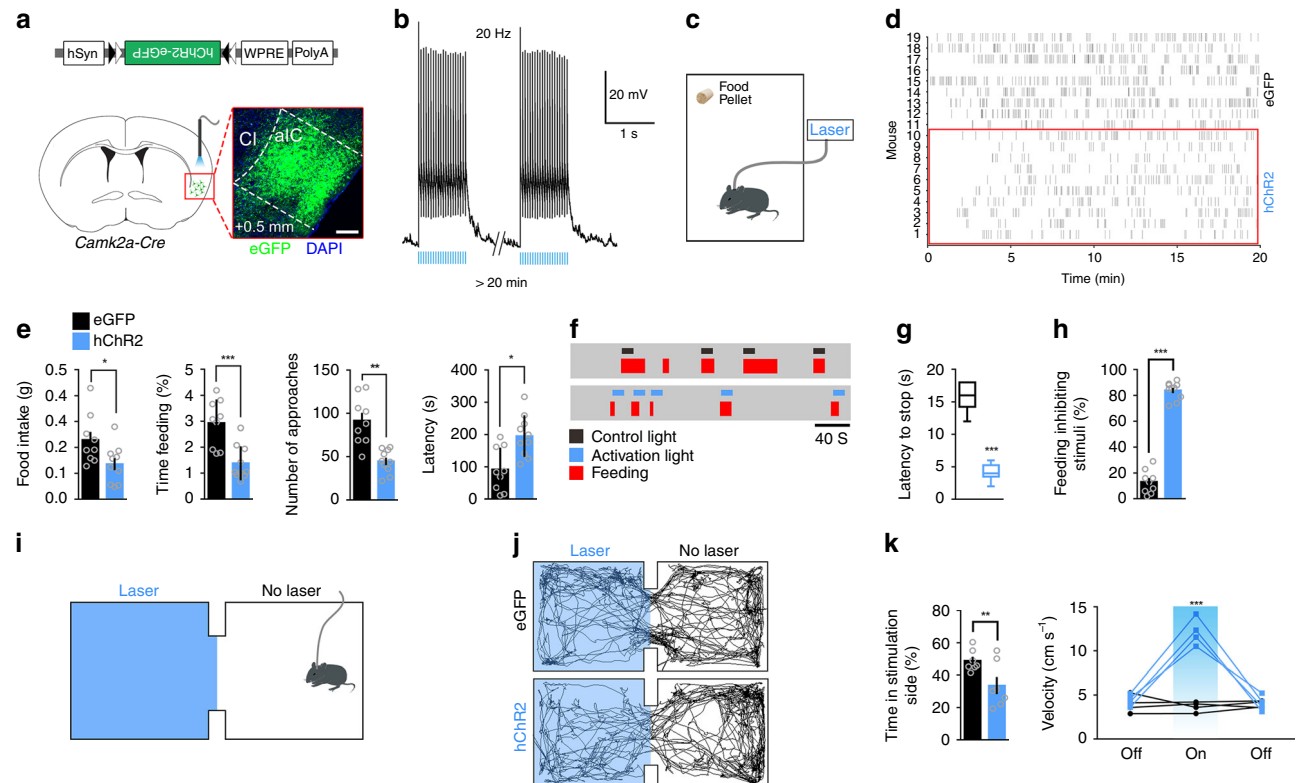

**Fig. 2 Activation of the right aIC^CamKII neurons suppresses food intake.** Diagrams illustrating the injected virus (upper) and injection site in the caudal segment of the right aIC (lower). The inserted image shows the expression of hChR2 (H134R)-eGFP in the right aIC. WPRE woodchuck hepatitis virus post-transcriptional regulatory element; pA poly a; Cl claustrum. Scale bar, 500 μm. **b** Brain slice whole-cell patch-clamp recording traces showing action potentials triggered in the aIC^CamKII neurons expressing hChR2 by 20 Hz, 10 ms, 473 nm laser pulses. **c** Schema to assess food intake for optogenetics. **d** Raster plot of food approaches in 24-h fasted mice expressing hChR2(H134R)-eGFP in the right aIC. **e** Total food consumption, percentage of time spent feeding, number of approaches to food and latency of the first approach to food were measured during a 20-min test ($n = 9$ for eGFP mice, $n = 10$ for hChR2 mice, Two-tailed unpaired $t$ test). **f** Raster plots showing feeding processes by one 24-h fasted mouse in its home cage. Activation light (473 nm) or control light (589 nm) was triggered after feeding began. **g** Latency to stop feeding in response to light activation (473 nm), 30 trails from five animals ($n = 5$ mice per group, Two-tailed unpaired $t$ test). **h** The percentage of light stimuli (473 nm) that inhibits feeding ($n = 9$ for eGFP mice, $n = 10$ for hChR2 mice, Two-tailed unpaired $t$ test). **i** Schema depicting the real-time place performance (RTPP) paradigm. **j** Representative locomotor trace of a mouse with the caudal segment of the right aIC^CamKII expressing eGFP or hChR2 that received the 20-Hz photostimulation in the laser compartment. **k** Percentage of time spent (left) and locomotor activity (right) in laser or no laser side ($n = 9$ for eGFP mice; $n = 10$ for hChR2 mice, Two-tailed unpaired $t$ test). *$P < 0.05$; **$P < 0.01$; ***$P < 0.005$. Data are presented as means ± SEM. Source data are provided as a Source Data file.

touching and consuming a food pellet, Two-tailed unpaired $t$ test, $t_{17} = 4.259$, $P = 0.005$) during a 20-min test (Fig. 2d, e). Moreover, photoactivation of these neurons decreased the number of approaches (the number of times at which the mice's nose touched the food pellet, two-tailed unpaired $t$ test, $t_{17} = 4.775$, $P = 0.0067$) and increased the latency of the first approach to food (two-tailed unpaired $t$ test, $t_{17} = 3.408$, $P = 0.0034$) (Fig. 2d, e). Activation of these neurons also interrupted ongoing feeding within a few seconds following the onset of photoactivation in 24-h fasted mice when mice were in their home cages (Fig. 2f–h, Two-tailed unpaired $t$ test, for latency to stop, $t_{20} = 10.7$, $P < 0.0001$; for the percentage of stimuli that inhibits feeding, $t_{17} = 19.64$, $P < 0.0001$). Thus, activation of the CamKII+ neurons in the caudal segment of the right aIC inhibited both the appetitive and consummatory phases of feeding behaviors.

We further tested whether activation of these neurons affects the valence of taste, cardiac function, or other classical consummatory behaviors and observed that optogenetic activation of these neurons has no effect on the mice bitter sensitivity (two-way ANOVA, $F_{3,56} = 0.6245$, $P = 0.6022$, Supplementary Fig. 5b), heart rate (Two-tailed unpaired $t$ test, $t_6 = 0.9329$, $P = 0.3869$, Supplementary Fig. 5c, d), drinking (two-tailed $t$ test, Water intake, $t_{12} = 0.1997$, $P = 0.8457$; Number of licks,

$t_{12} = 0.2915$, $P = 0.7756$; Time drinking, $t_{12} = 0.1993$, $P = 0.8454$, Supplementary Fig. 5e, f) or mating behaviors (two-tailed unpaired $t$ test, $t_{68} = 0.9092$, $P = 0.3702$, Supplementary Fig. 5g–i).

In the following, we tested the approach or avoidance behavior using real-time place preference assay (RTPP) and observed that mice expressing hChR2 in right aIC^CamKII neurons spent significantly less time on the photostimulated side and made more escape attempts (two-tailed unpaired $t$ test, Time in stimulation side, $t_{12} = 2.915$, $P = 0.0095$; Velocity, 1st light off step, $t_{17} = 0.754$, $P = 0.4772$, 1st light on step, $t_{17} = 10.23$, $P < 0.0001$, 2nd light off step, $t_{17} = 0.262$, $P = 0.802$, Fig. 2i–k). In contrast, optogenetic activation of these neurons induced no anxiety-like behaviors in two different assays of anxiety, including the elevated plus maze test (two-tailed unpaired $t$ test, Open arm entries, $t_{14} = 0.7477$, $P = 0.467$; Time in open arm, $t_{14} = 0.3227$, $P = 0.7517$, Supplementary Fig. 5j, k) and open field test (OFT) (two-tailed unpaired $t$ test, time in center, $t_{15} = 0.3872$, $P = 0.7041$; velocity, $t_{15} = 0.777$, $P = 0.4492$, Supplementary Fig. 5l, m).

To confirm the specificity of the caudal segment of the right aIC in mediating feeding behavior, we optogenetically activated the CamKII+ neurons in the rostral segment of the right aIC

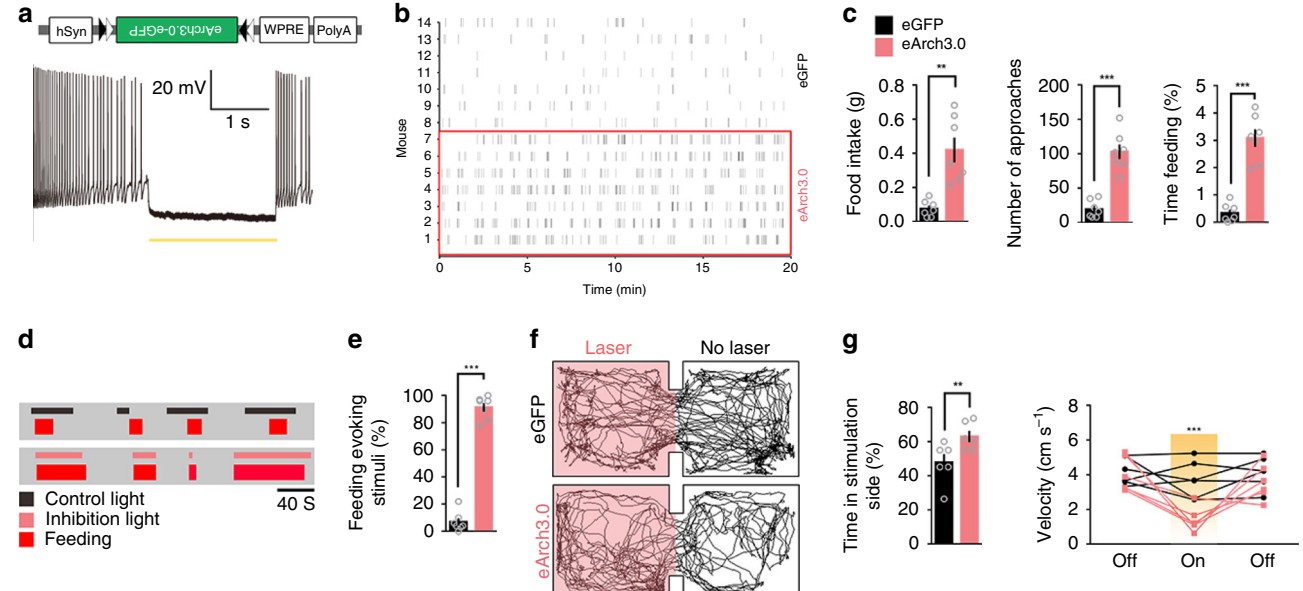

**Fig. 3 Inhibition of the right aIC^CamKII neurons promotes food intake. a** Diagrams illustrating the injected virus (upper) and sample slice recordings from the aIC^CamKII neurons expressing eArch3.0 (orange lines, 593-nm light) (lower). **b** Raster plot of feeding bouts in the fed mice expressing eArch3.0-eGFP in the caudal segment of the right aIC. **c** Total food consumption (left), number of approaches to food (middle) and the percentage of feeding time (right) were measured during a 20-min test ($n = 7$ mice for each group, Two-tailed unpaired $t$ test). **d** Raster plots showing feeding processes by one fed mouse in its home cage. Inhibition light (532 nm) and control light (644 nm) was triggered randomly. **e** The percentage of light stimuli (532 nm) that induces feeding ($n = 7$ mice for each group, Two-tailed unpaired $t$ test). **f** Representative locomotor trace of a mouse with the caudal segment of the right aIC^CamKII neurons expressing eGFP or eArch3.0 that received photoinhibition in the laser compartment. **g** Percentage of time spent (left) and locomotor activity (right) in laser or no laser sides ($n = 7$ mice per group, Two-tailed unpaired $t$ test). *$P < 0.05$; **$P < 0.01$; ***$P < 0.005$. Data are presented as means ± SEM. Source data are provided as a Source Data file.

(two-tailed unpaired $t$ test, $t_{12} = 0.569$, $P = 0.5799$, Supplementary Fig. 6a, b) or those in the right posterior IC (pIC) (two-tailed unpaired $t$ test, $t_{13} = 0.5801$, $P = 0.5718$, Supplementary Fig. 6c–e) and observed that photoactivation of the CamKII+ neurons in these two neighboring sites has no effect on feeding behavior.

**Silencing the right-side aIC^CamKII neurons promotes feeding**. We next asked whether inhibition of the CamKII+ neurons in the caudal segment of the right aIC increases food consumption under conditions in which motivation to eat is low. We silenced the right aIC^CamKII neurons expressing Cre-dependent AAV9-DIO-eArch3.0-eGFP virus with 532-nm laser light (Fig. 3a upper, Supplementary Fig. 7a). Brain slice recordings confirmed that aIC^CamKII neurons expressing eArch3.0 could be strongly inhibited by 532-nm laser light (Fig. 3a lower). Indeed, optogenetic silencing of the CamKII+ neurons in the caudal segment of the right aIC increased the total amount of food intake (Two-tailed unpaired $t$ test, $t_{12} = 3.758$, $P = 0.049$), the number of approach (Two-tailed unpaired $t$ test, $t_{12} = 7.167$, $P < 0.0001$) and the percentage of time spent feeding (Two-tailed unpaired $t$ test, $t_{12} = 7.995$, $P < 0.0001$) in fed mice expressing eArch3.0, but not in mice expressing eGFP (Fig. 3b, c). Moreover, silencing these neurons evoked feeding behavior in fed mice when the mice were in their home cages (Two-tailed unpaired $t$ test, $t_{12} = 20.14$, $P < 0.0001$, Fig. 3d, e). These data suggest that inhibition of neural activity within the caudal segment of right aIC is sufficient to induce feeding behavior.

In the RTPP assay, the mice expressing eArch3.0 in these neurons exhibited a significant preference for the photoinhibition-paired chamber (Fig. 3f–g, Two-tailed unpaired $t$ test, Time in stimulation side, $t_{12} = 3.915$, $P = 0.0083$; Velocity, 1st light off step, $t_{12} = 0.3843$, $P = 0.7097$, 1st light on step, $t_{12} = 0.676$, $P = 0.515$, 2nd light off step, $t_{12} = 0.6583$, $P = 0.5268$). In

contrast, drinking (two-tailed unpaired $t$ test, $t$ test, water intake, $t_{12} = 1.96$, $P = 0.0737$; number of licks, $t_{12} = 1.503$, $P = 0.1587$; time drinking, $t_{12} = 0.1523$, $P = 0.1523$, Supplementary Fig. 7b), mating (two-tailed unpaired $t$ test, $t$ test, $t_{34} = 1.335$, $P = 0.1908$, Supplementary Fig. 7c), and anxiety-like behaviors (Supplementary Fig. 7d–g, Two-tailed unpaired $t$ test, for Supplementary Fig. 7e, open arm entries, $t_{12} = 0.6325$, $P = 0.539$; time in open arm, $t_{12} = 0.874$, $P = 0.3993$, for Supplementary Fig. 7g, time in center, $t_{12} = 1.037$, $P = 0.3202$; velocity, $t_{12} = 1.078$, $P = 0.3022$) were unaffected. Taken together, these results indicate that the inhibition of the CamKII+ neurons in the caudal segment of the right aIC promotes food consumption in the absence of homeostatic deficit and at the same time induces approach behavior.

**Activating the left aIC^CamKII neurons induces no feeding**. We also explored whether the caudal segment of the left aIC^CamKII neuron activity is involved in mediating feeding or avoidance behavior. AAV virus carrying hChR2 was injected unilaterally in the caudal segment of the left aIC of *Camk2a-Cre* mice (Fig. 4a, Supplementary Fig. 8a). Optogenetic activation of these neurons in vivo at 20 Hz had no effect on feeding behavior in 24-h-fasted mice (two-tailed unpaired $t$ test, food intake, $t_{12} = 0.3688$, $P = 0.7187$; number of approaches, $t_{12} = 0.9881$, $P = 0.3426$; time feeding, $t_{12} = 0.2507$, $P = 0.8063$, Fig. 4b, c). Moreover, activation of these neurons produced no conditioned place aversion (Two-tailed unpaired $t$ test, time in stimulation side, $t_{12} = 0.4276$, $P = 0.6765$; velocity, 1st light off step, $t_{12} = 0.493$, $P = 0.631$, 1st light on step, $t_{12} = 0.8914$, $P = 0.3902$, 2nd light off step, $t_{12} = 0.5257$, $P = 0.6087$, Fig. 4d, e), induced no drinking (Two-tailed unpaired $t$ test, water intake, $t_{12} = 0.1104$, $P = 0.9139$; number of licks, $t_{12} = 0.5958$, $P = 0.5624$; time drinking, $t_{12} = 0.3115$, $P = 0.7607$, Supplementary Fig. 8b) or mating

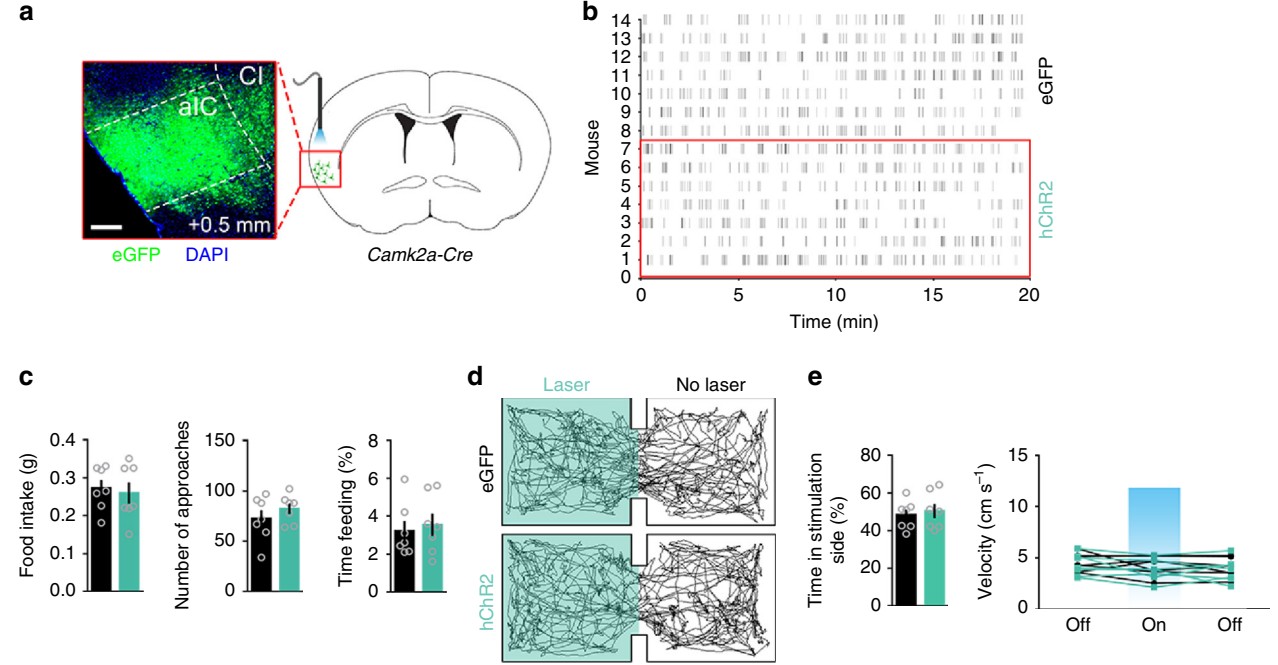

**Fig. 4 Activation of the left aIC^CamKII neuron has no effect on feeding behavior. a** The inserted image shows the expression of hChR2-eGFP in the caudal segment of the left aIC. Scale bar, 250 μm. **b** Raster plot of approaches to food in 24-h fasted mice expressing hChR2-eGFP in the caudal segment of the left aIC and receiving 20 Hz photostimulation. **c** Total food consumption (left), number of approaches (middle) and the percentage of feeding time (right) were measured (n = 7 mice per group, Two-tailed unpaired t test). **d** Representative locomotor trace of a mouse with the caudal segment of the left aIC^CamKII neurons expressing eGFP or hChR2 and receiving 20 Hz photostimulation in the laser compartment. **e** Percentage of time spent in the photostimulation side (left) and locomotor activity (right) in laser or no laser sides (n = 7 mice per group, Two-tailed unpaired t test). Data are presented as means ± SEM. Source data are provided as a Source Data file.

behavior (two-tailed unpaired t test, $t_{27} = 0.8559$, $P = 0.3996$, Supplementary Fig. 8c), and exerted no effect on anxiety-like behavior (Supplementary Fig. 8d–g; two-tailed unpaired t test, for Supplementary Fig. 8d, $t_{27} = 0.8559$, $P = 0.3996$; for Supplementary Fig. 8e, open arm entries, $t_{12} = 0.3174$, $P = 0.7564$; time in open arm, $t_{12} = 0.1824$, $P = 0.8583$; for Supplementary Fig. 8g, time in center, $t_{12} = 0.0082$, $P = 0.9936$; velocity, $t_{12} = 0.0059$, $P = 0.9953$). These data suggest that the caudal segment of the left aIC is unnecessary for feeding or avoidance behavior.

**Inhibiting the right aIC^CamKII neurons reverses anorexia.** To test the long-term effect of the right aIC^CamKII neuron activity, we injected an excitatory virus AAV9-DIO-hM3Dq-mCherry or inhibitory virus AAV9-DIO-hM4Di-mCherry in the caudal segment of the right aIC of *Camk2a-Cre* mice (Supplementary Fig. 9a, b). Electrophysiological analysis in acute slices confirmed that Clozapine N-oxide (CNO, 5 μM) activated spiking in neurons expressing hM3Dq (Fig. 5a, b). CNO treatment rapidly and markedly inhibited food intake in 24 h-fasted mice expressing hM3Dq in the caudal segment of the right aIC^CamKII neurons during a 1-h test (Two-way ANOVA, food intake, $F_{1,18} = 43.11$, $P < 0.0001$; feeding time, $F_{1,18} = 32.82$, $P < 0.0001$, Fig. 5c). Moreover, long-term CNO treatment significantly reduced body weight in mice expressing hM3Dq in these neurons but not in mice expressing mCherry. Body weight returned to normal after cessation of CNO treatment (two-way RM ANOVA, interaction: $F_{19,114} = 10.6$, $P < 0.0001$, Fig. 5d).

Electrophysiological analysis in acute slices also confirmed that CNO (5 μM) inhibited spiking in neurons expressing hM4Di (Fig. 5e, f). Of interest, administration of CNO (i.p.) to inhibit the CamKII+ neurons in the caudal segment of the right aIC had no effect on CCK-induced reduction of food intake in fasted mice (two-way ANOVA, saline, $F_{3,26} = 1.113$, $P = 0.3619$; CCK,

$F_{3,26} = 2.238$, $P = 0.1076$, Fig. 5g). However, chemogenetic inhibition of these neurons prevented LiCl-induced and LPS-induced reduction of food intake (two-way ANOVA, LiCl, $F_{3,26} = 3.287$, $P = 0.0365$; LPS, $F_{3,26} = 9.614$, $P = 0.0002$, Fig. 5g). Administration of saline (i.p.) has no effect on CCK-induced, LiCl-induced, or LPS-induced reduction of food intake (two-way ANOVA, Saline, $F_{3,26} = 0.4549$, $P = 0.7161$; CCK, $F_{3,26} = 0.4727$, $P = 0.7039$; LiCl, $F_{3,26} = 0.7194$, $P = 0.5394$; LPS, $F_{3,26} = 0.6813$, $P = 0.5715$, Supplementary Fig. 9c). Moreover, long-term CNO treatment significantly increased the body weight of mice expressing hM4Di in the CamKII+ neurons in the caudal segment of the right aIC, but not mice-expressing mCherry, and this effect was reversed post CNO injection (two-way RM ANOVA, interaction: $F_{18,126} = 9.078$, $P < 0.0001$, Bonferroni post hoc analysis mCherry with hM4Di on days 6–8 as indicated, Fig. 5h). Long-term Cisplatin treatment (every 2 days for 14 days) significantly reduced the body weight of the injected mice. Chronic chemogenetic inhibition of the right aIC^CamKII neurons significantly prevented further weight loss, while inhibition of the left aIC^CamKII neurons failed to rescue body weight loss (two-way RM ANOVA, interaction: $F_{16,64} = 57.1$, $P < 0.0001$, Bonferroni post hoc analysis mCherry with hM4Di on days 9–14 as indicated, Fig. 5i), implying that the caudal segment of the right aIC is critical in mediating feeding in response to aversive stimuli.

**The aIC^CamKII neurons directly innervate LH^vGluT2 neurons.** To elucidate the neurocircuitry involved in the aIC-mediated feeding behavior, we anatomically mapped the long-range projections of CamKII+ neurons in the aIC. First, we selectively expressed a Cre-dependent fluorophore AAV8-DIO-eGFP into the right aIC of *Camk2a-Cre* mice. Imaging showed that the aIC^CamKII neurons projected heavily to the hypothalamus (Supplementary Fig. 10a, b). It is of interest to note that, in the

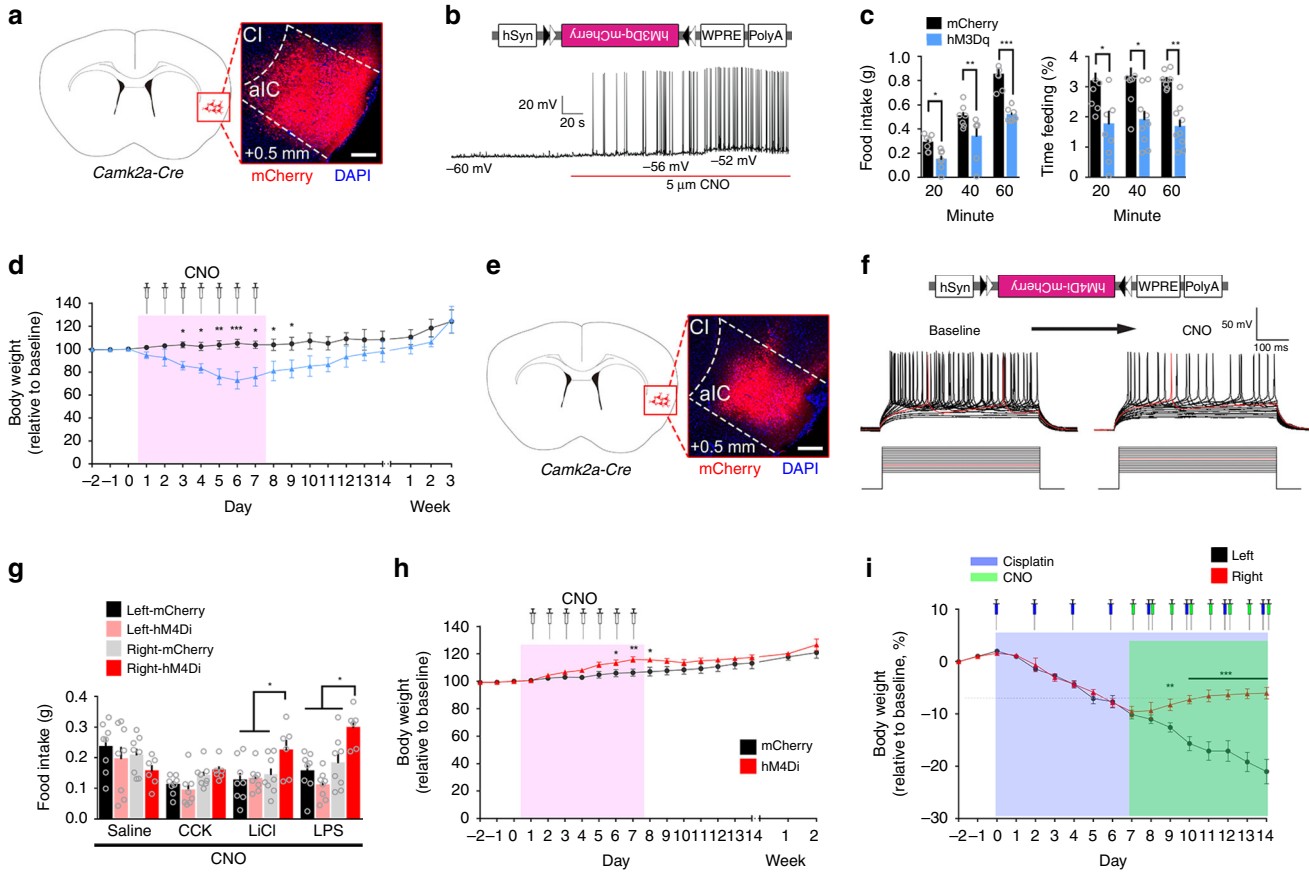

**Fig. 5 Inhibition of the right aIC^CamKII neurons rescues Cisplatin-induced anorexia. a** The inserted image shows the expression of hM3Dq-mCherry in the caudal segment of the right aIC. Scale bar, 250 μm. **b** Diagrams illustrating the injected virus (upper) and example current-clamp traces from the brain slice with the right aIC^CamKII neurons expressing hM3Dq before and after treatment with CNO. **c** Food intake (left) and percentage of feeding time (right) in CNO-treated 24 h-fasted mice expressing hM3Dq in the right aIC^CamKII neurons during 20, 40, or 60-min test ($n = 7$ mice for per group). **d** Chronic administration of CNO (pink area, 1 mg/kg, every 12 h for 7 days, i.p.) reduces body weight ($n = 8$ mice per group). **e** The inserted image shows the expression of hM3Di-mCherry in the caudal segment of the right aIC. Scale bar, 250 μm. **f** Diagrams illustrating the injected virus (upper) and current–voltage relationship of a representative aIC^CamKII neuron recorded before and after perfusion with CNO. Raw traces show individual voltage responses to a series of 500 ms current pulses from 100 to 200 pA with 10-pA steps. Red traces indicate the minimal current to induce action potentials. **g** Food intake in 24 h-fasted mice after administration of different anorexigenic agents (CCK (5 μg/kg), LiCl (150 mg/kg), and LPS (0.1 mg/kg) and CNO ($n = 8$ for left-mCherry, left-hM4Di, and right-mCherry group, $n = 6$ for right-hM4Di group). **h** Chronic administration of CNO (pink area, i.p.) increases body weight ($n = 9$ mice for control group, $n = 8$ mice for hM4Di group). **i** Chronic administration of Cisplatin (blue area, 4 mg/kg, i.p.) every 2 days for 14 days followed by CNO injection (green area, i.p.) every 12 h for 7 days after 7 days Cisplatin injection. Body weight of mice was measured every day ($n = 8$ mice per group). Two-way ANOVA with Bonferroni post hoc analysis. *$P < 0.05$; **$P < 0.01$; ***$P < 0.005$. Data are presented as means ± SEM. Source data are provided as a Source Data file.

hypothalamus, fiber terminals originating from the aIC were predominantly localized within the LH (Fig. 6a), a critical neuroanatomical substrate for feeding behavior[33]. Next, we designed a series of experiments to test whether the aIC^CamKII neurons have direct projections to the LH. Immunohistochemical staining showed that stimulation of the projection terminals with 473-nm laser induced significant Fos expression in the entire length of the anterior–posterior (A–P) axis of the LH, but a greater number between 1.06 and 1.58 mm posterior of Bregma (Two-way ANOVA, Fos expression at $-0.7$ mm, $F_{2,28} = 1.364$, $P = 0.5503$; Fos expression at $-1.06$ mm, $F_{2,28} = 2.871$, $P = 0.073$; Fos expression at $-1.34$ mm, $F_{2,28} = 43.47$, $P < 0.0001$; Fos expression at $-1.58$ mm, $F_{2,28} = 5.581$, $P = 0.0031$; Fos expression at $-1.94$ mm, $F_{2,28} = 3.554$, $P = 0.041$, Fig. 6b), indicating that terminals originating from the aIC form functional synaptic connections with the neurons in the LH. To characterize the aIC to LH connection, whole-cell electrophysiological recording was used to examine light-evoked inhibitory postsynaptic currents (IPSCs) and excitatory postsynaptic currents (EPSCs) from LH

neurons in brain slices. This was accomplished by alternation between the holding potential of 0 mV (IPSCs) and $-70$ mV (EPSCs) (Fig. 6c). Light activation resulted in eEPSCs and no eIPSCs in all cells tested. Moreover, evoked EPSCs were blocked by the glutamatergic receptor antagonist CNQX (Fig. 6d), indicating that the terminals originating from the aIC formed functional glutamatergic synaptic connections with the neurons in the LH. To further test the cell-type specificity of this projection, we performed Cre-dependent, rabies virus-based retrograde tracing from vGluT2 neurons of the LH. We injected synaptic retrograde ΔG-rabies viruses encoding mCh (RV) into the LH of *Slc17a6* (encodes vesicular glutamate transporter, vGluT2)-*ires-Cre* mice. Positive signals were detected in the aIC and colocalized well with CamKII (Fig. 6e, f), which confirms that the aIC^CamKII neurons have direct projection to the vGluT2+ neurons in the LH.

To confirm that the aIC receives input from thalamus, we further injected retrograde cholera toxin subunit B (CTB, conjugated with Alexa-555) into the right aIC and observed significant signals in the area VPMpc (Supplementary Fig. 10c).

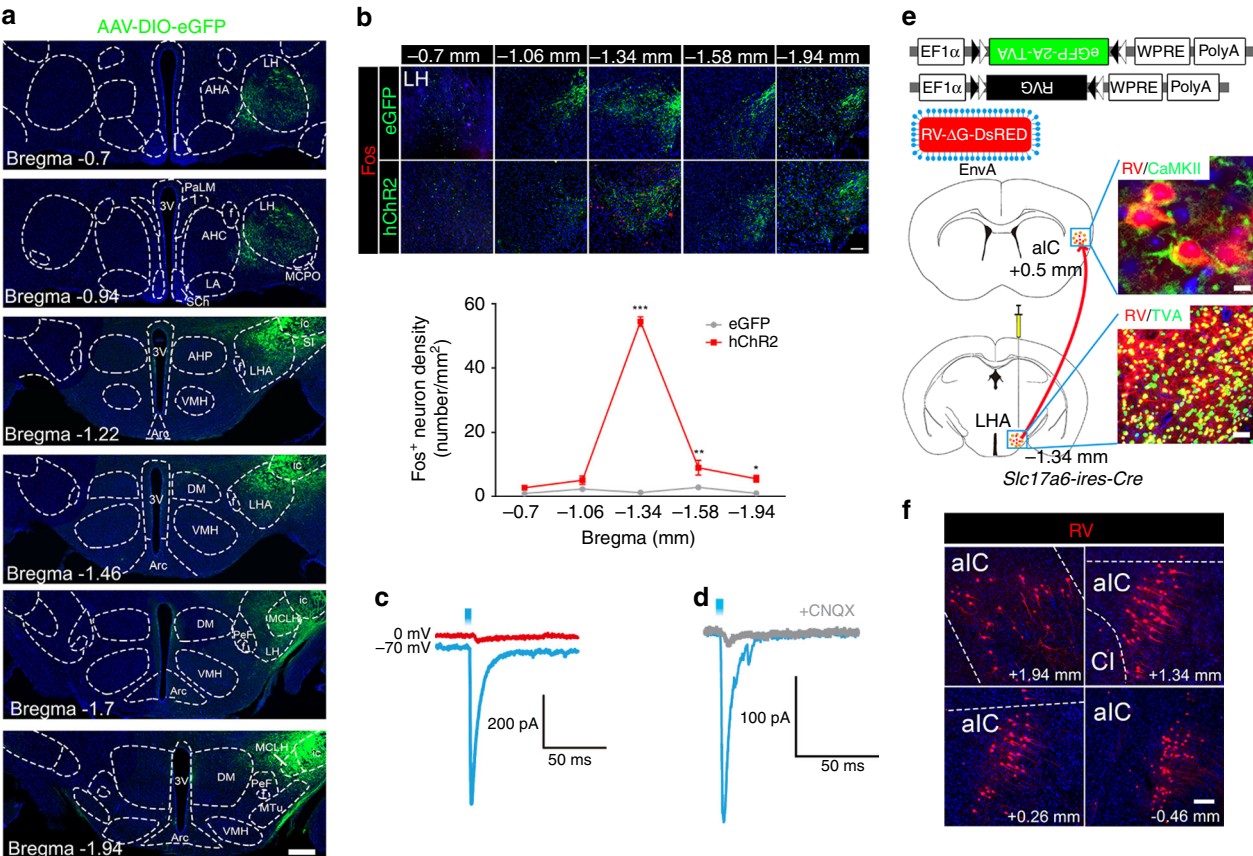

**Fig. 6 Right aIC<sup>CamKII</sup> neurons project to the LH directly.** a Representative histology of images from the LH showing CamKII+ fiber terminal in green after injecting hSyn-DIO-eGFP into the right aIC of *Camk2a-Cre* mice. Distribution of the fiber terminals in the area of hypothalamus is shown from bregma −0.70 mm to bregma −1.94 mm. Scale bar, 250 μm. AHA anterior hypothalamic area, anterior part; AHC anterior hypothalamic area, central part; Arc arcuate hypothalamic nucleus; f fornix; LA lateroanterior hypothalamic nucleus; 3V 3rd ventricle; ic internal capsule; SI substantia innominate; AHP anterior hypothalamic area, posterior part; MTu medial tuberal nucleus; VMH ventromedial hypothalamic nucleus; MCLH magnocellular nucleus of the lateral hypothalamic; DM dorsomedial hypothalamic nucleus; PeF perifornical nucleus; PaLM paraventricular hypothalamic nucleus, lateral magnocellular part; SCh suprachiasmatic nucleus; MCPO magnocellular preoptic nucleus. b eGFP or hChR2-eGFP injected in the right aIC and the terminals in LH were activated by light and immunostained with Fos (red) 30 min after light stimulation (n = 5 mice per group, Two-way ANOVA with Bonferroni post hoc analysis). Scale bar, 200 μm. c Light can evoke EPSCs (held at −70 mV), but not IPSCs (held at 0 mV). n = 6 cells. d Single light evoked EPSC, which can be blocked by CNQX. n = 5 cells. e Diagram illustrating the injected virus (AAV-FLEX-eGFP-2A-TVA, AAV-FLEX-RVG, and ΔRVG-DsRed) (upper) and injection sites (lower) for rabies-based projection-specific monosynaptic tracing of inputs to LH<sup>vGluT2</sup> neurons in *Slc17a6-ires-Cre* mice. Rabies are colocalized with CamKII in the right aIC (upper). Neurons co-infected (starter cell) with helper AAV and rabies are shown in yellow. f Distribution of monosynaptic input (Rabies-DsRed neurons) from the right aIC to the LH<sup>vGluT2</sup>. Scale bar, 100 μm (Right upper). Scale bar, 25 μm (Right lower). Source data are provided as a Source Data file.

Results from in situ hybridization of *Slc17a6* mRNA showed that 94.5% of CTB555-labeled neurons overlapped with *Slc17a6* mRNA, indicating that VPMpc-to-aIC projecting neurons are primarily glutamatergic (Supplementary Fig. 10c).

**The right aIC-to-LH projections mediate feeding suppression.**
To assess the functional contribution of the right aIC<sup>CamKII</sup>-to-LH projection to feeding behavior, we introduced hChR2-eGFP into the caudal segment of the right aIC of *Camk2a-Cre* mice (Fig. 7a and Supplementary Fig. 11a). Photoactivation of the right aIC<sup>CamKII</sup> neuronal terminals in LH significantly reduced the amount of food intake, the number of approaches, and the percentage of time spent feeding in the fasted mice (two-tailed unpaired $t$ test, food intake, $t_{20} = 2.445$, $P = 0.0239$; number of approaches, $t_{20} = 5.451$, $P < 0.0001$; time feeding, $t_{20} = 5.814$, $P < 0.0001$, Fig. 7b, c) and also inhibited ongoing feeding when mice were in their home cages (two-tailed unpaired $t$ test, latency to stop, $t_{20} = 15.33$, $P < 0.0001$; feeding-inhibited stimulation,

$t_{20} = 14.31$, $P < 0.0001$, Fig. 7d, e). In addition, activation of the right aIC<sup>CamKII</sup>-to-LH pathway produced place avoidance (two-tailed unpaired $t$ test, time in stimulation side, $t_{20} = 3.493$, $P = 0.0023$; velocity, 1st light off step, $t_{20} = 0.5406$, $P = 0.5948$; 1st light on step, $t_{20} = 5.778$, $P < 0.0001$, 2nd light off step, $t_{20} = 0.7825$, $P = 0.3862$, Fig. 7f, g), while it had no effects on drinking (two-tailed unpaired $t$ test, water intake, $t_{20} = 0.523$, $P = 0.6067$; number of licks, $t_{20} = 0.5141$, $P = 0.5942$; time drinking, $t_{20} = 0.1318$, $P = 0.8965$, Supplementary Fig. 11b) or mating behavior (two-tailed unpaired $t$ test, $t_{26} = 0.1827$, $P = 0.8565$, Supplementary Fig. 11c).

To confirm this observation, we used another labeling strategy. AAV8-hSyn-DIO-hChR2-eGFP virus was injected into the right aIC after a retrogradely transported virus canine adenovirus-2 encoding Cre-recombinase (Cav2-Cre) was injected into the LH (Supplementary Fig. 12a, b). This resulted in the specific hChR2 expression in the right aIC neurons that projected to LH. Photoactivation of these neurons inhibited food consumption in 24 h-fasted mice (Two-tailed unpaired $t$ test, food intake,

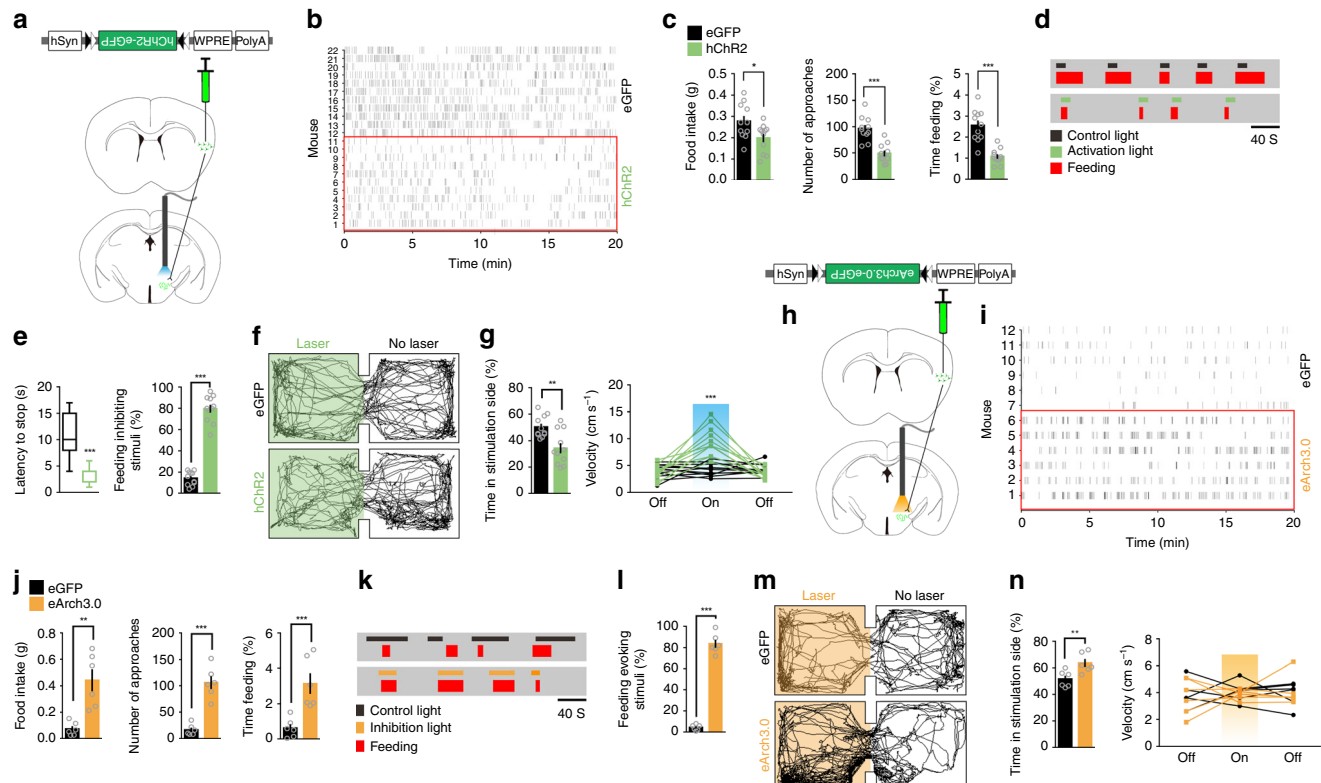

**Fig. 7 Right aIC$^{CamKII}$-to-LH projections mediate feeding and avoidance behaviors. a** Diagrams illustrating *Camk2a-Cre* mice injected with hChR2-eGFP in the right aIC and photostimulated in the LH. **b** Raster plot of approaches to food in 24-h-fasted mice with the right aIC$^{CamKII}$-to-LH terminals being activated. **c** Activation of the right aIC$^{CamKII}$-to-LH terminals inhibits food intake in 24-h-fasted mice. Total food consumption, number of approaches and the percentage of feeding time were measured ($n = 11$ mice per group). **d** Raster plots showing feeding processes by one 24-h-fasted animal in its home cage. Activation light or control light was triggered after feeding began. **e** Latency to stop feeding (left) in response to light activation and the percentage of light stimuli that inhibits feeding ($n = 11$ mice per group). **f** Representative locomotor trace of a mouse with the right aIC$^{CamKII}$ neurons expressing eGFP or hChR2 and the right LH receiving 20-Hz photostimulation in the laser compartment. **g** Percentage of time spent in the photostimulation side (left) and locomotor activity in laser or no laser sides (right) ($n = 11$ mice per group). **h** Diagrams illustrating *Camk2a-Cre* mice with eArch3.0-eGFP injected in the right aIC and photostimulation in the right LH. **i** Raster plot of feeding approach in fed mice with the right aIC$^{CamKII}$-to-LH terminals being photoinhibited. **j** Total food consumption, number of approaches to food, and the percentage of feeding time were measured ($n = 6$ mice per group). **k** Raster plots showing feeding processes by one fed animal in its home cage. Inhibition light or control light was triggered randomly. **l** The percentage of light stimulation that induces feeding ($n = 6$ mice per group). **m** Representative locomotor trace of a mouse with the right aIC$^{CamKII}$-to-LH terminals expressing eGFP or eArch3.0 and receiving photoinhibition in the laser compartment. **n** Percentage of time spent in the photoinhibition side and locomotor activity in laser or no laser sides ($n = 6$ mice per group). Two-tailed unpaired *t* test, *$P < 0.05$; **$P < 0.01$; ***$P < 0.005$. Data are presented as means ± SEM. Source data are provided as a Source Data file.

$t_8 = 2.803$, $P = 0.0230$; number of approach, $t_8 = 3.153$, $P = 0.0275$; time feeding, $t_8 = 4.786$, $P = 0.0014$, Supplementary Fig. 12c) and induced place avoidance (Two-tailed unpaired *t* test, time in stimulation side, $t_8 = 4.329$, $P = 0.025$; velocity, 1st light off step, $t_8 = 0.8011$, $P = 0.4462$ 1st light on step, $t_8 = 8.526$, $P < 0.0001$, 2nd light off step, $t_8 = 1.824$, $P = 0.1056$, Supplementary Fig. 12d, e), while having no effects on drinking behavior (Two-tailed unpaired *t* test, water intake, $t_8 = 0.7801$, $P = 0.4578$; number of licks, $t_8 = 1.206$, $P = 0.2623$; time drinking, $t_8 = 0.3376$, $P = 0.7443$, Supplementary Fig. 12f), consistent with the aforementioned finding that the right aIC$^{CamKII}$-to-LH pathway is sufficient to inhibit feeding and induce aversive effects.

In contrast, specifically silencing the right aIC$^{CamKII}$-to-LH projection terminals via introducing eArch3.0 in the right aIC$^{CamKII}$ neurons and stimulating projection terminals in the LH with 532-nm laser (Fig. 7h, Supplementary Fig. 13a) promoted food consumption in fed mice (Fig. 7i–l, Two-tailed unpaired *t* test, for Supplementary Fig. 7j, food intake, $t_{10} = 4.341$, $P = 0.0075$; number of approach, $t_{10} = 5.593$, $P < 0.0001$; time feeding, $t_{10} = 4.082$, $P = 0.0022$; for Supplementary Fig. 7l, $t_{10} = 19.9$, $P < 0.0001$), as well as place preference (Two-tailed

unpaired *t* test, time in stimulation side, $t_{20} = 3.493$, $P = 0.0023$; velocity, 1st light off step, $t_{20} = 0.5406$, $P = 0.5948$, 1st light on step, $t_{20} = 5.778$, $P < 0.0001$, 2nd light off step, $t_{20} = 0.7825$, $P = 0.3862$, Fig. 7m, n), while having no effects on drinking behavior (Two-tailed unpaired *t* test, water intake, $t_{10} = 1.253$, $P = 0.2386$; number of licks, $t_{10} = 1.949$, $P = 0.0798$; time drinking, $t_{10} = 0.2572$, $P = 0.8023$, Supplementary Fig. 13b). Taken together, these results indicate that the right aIC$^{CamKII}$-to-LH pathway mediates feeding suppression and place avoidance.

## Discussion

In this study, we investigated how aversive stimuli alter insular activity and how direct manipulation of neuronal activity in the aIC affects aversive stimuli-induced feeding responses. Combining virus tracing, optogenetics, pharmacogenetics, photometry, and behavioral studies, we demonstrated that the caudal segment of the right aIC responds to diverse aversive stimuli in vivo and mediates anorexia via innervation of the LH. Therefore, we have identified a neural circuit from the cortex to the hypothalamus

that interrupts the homeostatic system and mediates feeding responses under emergency conditions.

The insular cortex is known to process polymodal information about bodily states, including visceral[34], gustatory[35], somatosensory[36], and auditory modalities[37]. In this study, we identified a segment in the right aIC that gates feeding in response to aversive anorexigenic stimuli. Interestingly, when we compared it with the bitter center and sweet center identified by Zuker's group[38], we found that this segment is just located between these two taste centers. These data demonstrate topographic segregation in the functional architecture of the insula[39–41]. It is possible that inputs carrying information from outside and inside the body project to topographically organized insular sensory regions. More tracing work is required to carefully analyze the connection between the insula and the thalamus or the PBN, such as the neuronal types, subregion, and left–right difference that serves sensory function.

The LH is a critical neuroanatomical region that is tied to consummatory and motivational behaviors. vGAT and vGluT2-expressing LH neurons produce a bidirectional output signal, which is directly or indirectly conveyed to the ventral tegmental area (VTA) dopamine neurons to homeostatically invigorate behavioral output. Inhibitory GABAergic subcortical fibers that innervate LH may originate from the lateral septum[42] and much of the basal forebrain and the extended amygdala, including the nucleus accumbens shell[43,44], the bed nucleus of the stria terminalis (BNST)/preoptic area[45], the ventral pallidum[46] and nucleus basalis/substantia innominate[47]. However, the source of excitatory inputs to the LH is still unknown. In this study, we showed that the glutamatergic input from the aIC to the LH inhibits food intake, consistent with the previous finding that vGluT2-expressing LH neurons produce an output signal to inhibit feeding and induce place avoidance. Therefore, we propose a model that environmental representations, such as aversive visceral stimuli, are likely encoded in the right insular cortex, which in turn convey representational information to the LH neuronal circuits to regulate feeding and rewarding. Previous work has shown that the LH may exert its effect on feeding directly via projecting to the PBN[48,49] to suppress feeding behavior or indirectly to the lateral habenula (LHb)[50] or VTA[51] by affecting the rewarding aspect of food consumption. More work is required to identify downstream brain regions that elicit the feeding suppression function of the aIC-to-LH circuit.

Craig[6] has proposed laterality differences in interoceptive perception related to emotional processing for the insular cortex in humans. In this study, we found that the right but not the left aIC in mice is critical in sensing aversive visceral stimuli and controlling feeding behavior. Moreover, inactivation of the right but not the left aIC reverses chemotherapy-induced weight loss. Consistently, one previous study in rats has also reported lateralized CREB activation in conditioned taste aversion with the right IC being more obvious than the left IC[52]. These findings suggest that hemispheric functional lateralization of aIC is not unique to humans but may be present in rodents as well. Furthermore, our findings are reminiscent of the right hemispheric dominance in negative emotional processing and expression[53,54] and of the right aIC hemispheric dominance in interoceptive attention in humans[15,55–57]. Recently, more and more evidence in rodents indicates left–right asymmetry for cognitive and emotional faculties. Several brain regions have been reported to be lateralized for corresponding behavioral functions, such as anterior cingulate cortex (ACC) for observational fear[58,59], mPFC for anxiety[60], amygdala for pain and fear[61–64], hypothalamus for energy metabolism[65,66], and hippocampus for learning and spatial navigation[67–69], respectively. However, further evidence is required to understand brain left–right asymmetries at

microscopic levels involving molecules, synapses, neurons, and neuronal circuits. Our findings here may provide an entry point for addressing basic neuronal underpinnings of insular functions and malfunctions in an accessible animal model.

To conclude, in this study, we have identified a pathway from the right insular cortex to the right LH that controls the intake of food. Additionally, to avoid potential confounding effects caused by lateralization, care should be taken in future behavioral studies when examining the role of a circuit.

## Methods

**Animals.** Camk2a-Cre mice (B6.Cg-Tg(Camk2a-cre)$^{T29-1Stl}$/J, strain 005359, RRID: IMSR_JAX:005359) and Slc17a6-ires-Cre mice (B6J.129S6(FVB)-Slc17a6$^{tm2(cre)}$ $^{Lowl}$/MwarJ, strain 028863, RRID:IMSR_JAX:028863) were obtained from the Jackson Laboratory. The male mice were used for behavioral test, and both the male and female mice were used for tracing image. Camk2a-Cre, Slc17a6-ires-Cre and C57BL/6J mice at 2–4 months of age were housed on a 12-h light/dark cycle under standard conditions in the animal facility with food and water ad libitum unless otherwise noted.

All animal care and experimental procedures complied with all relevant ethical regulations, were strictly conducted in accordance with the Guidelines for the Care and Use of Laboratory Animals of Zhejiang University and were approved by the Institutional Animal Care and Use Committee at Zhejiang University.

**Regents.** CNO, LiCl, LPS, Cisplatin, CNQX, and DAPI dye were obtained from Sigma Aldrich. Rat Ghrelin and mouse cholecysokinin (CCK) were obtained from Tocris Bioscience. CTB, Alexa Fluor$^{TM}$ 555 were obtained from Thermo Fisher.

**Viral constructs.** The following AAV viruses were provide by Shanghai Taitool Bioscience Co., Ltd: AAV9-hSyn-DIO-hM3Dq-mCherry, AAV9-hSyn-DIO-hM4Di-mCherry, AAV8-hSyn-DIO-mCherry, AAV8-hSyn-DIO-eGFP, AAV9-CamKIIα-mCherry, AAV8-hSyn-DIO-hChR2(H134R)-eGFP, AAV8-CamKIIα-hChR2(H134R)-eGFP, AAV9-hSyn-DIO-eArch3.0-eGFP, AAV9-mCamKIIα-GCaMP6f, AAV9-hSyn-DIO-mCherry, and Cav2-CMV-Cre. AAV9-CAG-DIO-TVA-EGFP, AAV9-CAG-DIO-RG and EnvA-pseudotyped, glycoprotein (RG)-deleted and DsRed-expressing rabies virus (RV-EvnA-DsRed, RV) were provided by Wuhan BrainVTA Co., Ltd.

**Stereotaxic surgery.** 2–4 months old mice were deeply anesthetized with 5% isoflurane (vol per vol) in oxygen and then kept with 1.5% isoflurane. Surgery was performed with a stereotaxic frame (Stoelting). For caudal segment of aIC injection, 150 nl of virus was injected unilaterally on site (+0.5 mm antero-posterior (AP); ±3.85 mm medio-lateral (ML); −2.82 mm dorso-ventral (DV) relative to Bregma). For right posterior insular cortex (pIC), virus was injected unilaterally on site (−1.10 mm AP; −3.90 mm ML; −3.25 mm DV relative to Bregma). For rostral segment of the right aIC, virus was injected unilaterally on site (+1.6 mm AP; −3.1 mm ML; −1.8 mm DV relative to Bregma). Virus was injected into each location at 0.03 μl/min. The syringe was not removed until 15–20 min after the end of infusion to allow diffusion of the virus. After injection, mice were single housed and allowed 2–4 weeks for viral expression and recovery from surgery.

For soma photostimulation or photoinhibition experiments, recombinant AAVs expressing hChR2 (H134R), eArch3.0, or eGFP were unilaterally injected into the aIC of Camk2a-Cre mice or wild-type mice. Three weeks after surgery, a fibreoptic cannula with 0.22 NA or 0.37 NA was implanted above the injected aIC (−2.67 DV) and mice were allowed another 1 week for recovery from surgery before behavioral test.

For axon terminal photostimulation experiments, recombinant AAVs expressing hChR2(H134R), eArch3.0, or eGFP were unilaterally injected into the aIC of Camk2a-Cre mice. Three weeks after surgery, a fibreoptic cannula with 0.22 NA was implanted above the LH (−1.28 mm AP; ±1.23 mm ML, −5.28 mm DV). Mice were allowed another 1 week for recovery from surgery before behavioral test.

For chemogenetic stimulation or inhibition, recombinant AAVs expressing hM3Dq, hM4Di, or mCherry were injected into the caudal segment of the aIC of Camk2a-Cre mice unilaterally. Four weeks later, the mice were received an i.p. injection of CNO (1 mg/kg) or the equivalent volume of saline and were allowed to recover in their home cage for 30 min before behavioral test. The mice with virus distributed mainly from Bregma +1.34 mm to Bregma +0.02 mm were used to analyze the related behaviors.

For photometry experiments, recombinant AAVs expressing mCamKIIα-GCaMP6f were injected unilaterally into the aIC of wild-type mice. Two weeks later, a fibreoptic cannula with 0.37 NA was implanted above the injected aIC (−2.67 DV). Mice were allowed 1 week for recovery from surgery before behavioral test.

For projection-mapping experiments, recombinant AAVs expressing GFP were injected unilaterally into the aIC of Camk2a-Cre mice. Eight weeks later, mice were sacrificed, fixed, and sections were collected for imaging.

For monosynaptic-tracing experiments, *Slc17a6-ires-Cre* mice were microinjected unilaterally in the LH with 150 nl viral cocktail (1:1) with AAV-CAG-DIO-TVA-EGFP to allow the initial infection of LH starter neurons. AAV-CAG-DIO-RG coding for the rabies virus envelope glycoprotein was also injected into the LH at the same time to allow the trans-synaptic spread of virus. Three weeks later, 200 nl of the modified rabies virus was microinjected into the same location. Another one week later, mice were sacrificed and fixed. Brain sections were collected for imaging.

**Immunohistochemistry**. Mice were sacrificed and perfused with 4% PFA in PBS. Brains were fixed overnight in PFA, cryoprotected in 30% sucrose (wt per vol) at 4 °C for 24 h, and then coronally sectioned at 40 μm on a cryostat (Leica CM1900). Brain sections were blocked with 10% NGS and 0.3% Triton-X in PBS for 1 h at room temperature, and then incubated with primary antibody (rabbit anti-Fos, SYSY system, 226 003, 1:10,000, RRID:AB_2231974; mouse anti-CamKII, Abcam, ab22609, 1:500, RRID:AB_447192; mouse anti-GAD67, Millipore, MAB5406, 1:1,000, RRID:AB_2278725) at 4 °C for 48 h. Sections were then washed with PBS for three times and incubated with secondary antibody (Alexa Fluor 488 goat anti-rabbit, Life Technologies, a11008, 1:2000, RRID:AB_143165; Alexa Fluor 555 donkey anti-rabbit, Life Technologies, a10040, 1:2000, RRID:AB_2534016; Alexa Fluor 488 goat anti-mouse, Life Technologies, a11029, 1:2000, RRID:AB_138404) for 2 h at room temperature. Sections were mounted and imaged with an Olympus VS120 microscope. For detection of Fos in ChR2 or GFP control mice, animals received 10-min of 20-Hz, ~5 mW photostimulation (488 nm) and were perfused 30 min later. For detection of Fos in mice with different aversive stimuli, mice were perfused 1.5 h after injection (i.p.) with different regents or electric foot shock unless otherwise mentioned.

**RNAscope in situ hybridization**. Animals were anesthetized and perfused with DEPC-PBS followed by ice-cold 4% PFA in DEPC-PBS. Brains were dissected and post-fixed over night at 4 °C and dehydrated with 30% sucrose in DPEC-PBS. Afterward, brains were sectioned at 20 μm thickness and mounted directly onto glass slides. RNAscope assay were performed according to RNAscope® Multiplex Fluorescent Reagent Kit v2 User Manual (ACD Bio). Probe against *Slc17a6* (vGluT2) mRNA was obtained from ACD Bio (Mm-*Slc17a6*−319171). Images were captured with an Olympus VS120 microscope.

**Acute brain-slice preparation and electrophysiology**. The mouse was deeply anesthetized with diethyl ether and the brain was rapidly removed and placed in ice-cold, high-sucrose cutting solution containing (in mM): 194 sucrose, 30 NaCl, 26 NaHCO3, 10 glucose, 4.5 KCl, 1.2 NaH2PO4, 7 MgSO4, 0.2 CaCl2, and 2 MgCl2. Slices were cut on a Leica vibratome (VT1200S) in the high-sucrose cutting solution and immediately transferred to an incubation chamber with artificial cerebrospinal fluid (ACSF) containing (in mM) 119 NaCl, 26.2 NaHCO3, 11 glucose, 2.5 KCl, 1 NaH2PO4, 1.3 MgCl2, and 2.5 CaCl2. The slices were allowed to recover at 34 °C for 30 min before being allowed to equilibrate at room temperature for another hour. During recordings, the slices were placed in a recording chamber constantly perfused with warmed ACSF (28–30 °C) and gassed continuously with 95% O2 and 5% CO2. For whole-cell voltage-clamp recording, the recording pipettes (3–5 MΩ) were filled with a Cs-based low Cl− internal solution containing (in mM): 135 CsMeSO3, 10 HEPES, 1 EGTA, 3.3 QX-314, 4Mg-ATP, 0.3 Na-GTP, 8 Na2-phosphocreatine, 290 mOsm kg−1, adjusted to pH 7.3 with CsOH. For whole-cell current-clamp recording, the recording pipettes were filled with (in mM) 110 K-gluconate, 40 KCl, 10 HEPES, 2 Mg-ATP, 0.5 Na2-GTP, and 0.2 EGTA 290 mOsm kg−1, adjusted to pH 7.3 with KOH. Data were digitized at 10 kHz, collected with a MultiClamp 700B amplifier, and analyzed by pClamp10 software (Molecular Devices, Sunnyvale, USA). Membrane potential was held at −70 mV to record AMPAR-mediated current and at 0 mV to record GABAA receptor-mediated IPSC. The following drugs were used diluted in ACSF as indicated: 100 μM Picrotoxin (PTX) and 20 μM CNQX. For confirmation of the efficiency of chemogenetic activation (hM3Dq) or silencing (hM4Di) of synaptic transmission, 5 μM of CNO diluted in ACSF was used. For ChR2 validation and circuit mapping, a blue light-emitting diode (NWEDOON) controlled by digital commands from the 1440 A was connected to the epifluorescence port of an Olympus BX51 microscope to deliver photostimulation. To record light-evoked EPSCs, 1–2 ms at λ = 473 nm and 1–10 mW/mm² blue light was delivered through the objective to illuminate the entire field of view. For optogenetic silencing validation, cells were activated using currents ranging from 0 to 45 pA (ΔI = 5 pA) for 3 s in duration injected under current clamp mode and photosilenced using a 532-nm LED light source (NEWDOON) sending constant light at 10 mW/mm².

**In vivo fiber photometry**. AAV-mCamKIIα-GCaMP6f was injected unilaterally into the aIC. Two weeks later, optic-fiber cannulas (fiber: core = 230 μm, NA = 0.37, NEWDOON) were implanted 0.15 mm above to the injection side. Behavioral test started one week later to allow for recovery from surgery. To record fluorescence signals, laser beam from a 488-nm laser (OBIS 488LS; Coherent) was reflected by a dichroic mirror (MD498; Thorlabs), focused by a ×10 objective lens (NA = 0.3; Olympus) and then coupled to an optical commutator (Doric Lenses). To record fluorescence signals, laser beam from a 488-nm laser (OBIS 488LS;

Coherent) was reflected by a dichroic mirror (MD498; Thorlabs), focused by a ×10 objective lens (NA = 0.3; Olympus) and then coupled to an optical commutator (Doric Lenses). An optical fiber (230 μm O.D., NA = 0.37, 2-m long) guided the light between the commutator and the implanted optical fiber. The laser power was adjusted at the tip of optical fiber to the low level of 0.01–0.02 mW to minimize bleaching. The GCaMP fluorescence was bandpass filtered (MF525-39, Thorlabs) and collected by a photomultiplier tube (R3896, Hamamatsu). An amplifier (C7319, Hamamatsu) was used to convert the photomultiplier tube current output to voltage signals, which was further filtered through a low-pass filter (40 Hz cut-off; Brownlee 440). The analog voltage signals were digitalized at 500 Hz and recorded by a Power 1401 digitizer.

For chemogenetic assay, mice were acclimated to the behavioral chamber (30 cm × 15 cm × 55 cm) for 30 min, and then injected intraperitoneally with LiCl (150 mg/kg) or Cisplatin (4 mg/kg). Calcium signals were recorded for another 30 min. Photometry setting, including laser power and time constant, were the same for every mouse and every recording session. Photometry data were subjected to minimal processing consisting of only within-trial fluorescence normalization.

**In vivo photostimulation**. For all optical activation studies, mice received blue light laser stimulation (473 nm, NEWDOON) of ~5 mW with a 10 ms pulse width. For optical inhibition studies, mice received constant green light stimulation (532 nm, NEWDOON) of ~10 mW. The lasers were triggered and pulses were controlled with Intelligent light system software.

**Animal behaviors**. All mice used in behavioral assays were allowed to recover from surgery of AAV injection and Ferrule fibers implantation for at least 4 weeks.

**Feeding behavior for optogenetic assay**. Before the test, mice were transferred into an empty testing cage in the behavioral testing room to habituate for at least 1 h. For 24 h-fasted feeding test, mice were food-deprived the day before test, with water provided ad libitum. The mice were then briefly anesthetized with isoflurane and coupled with optic fibers. 20 min after recovery, mice were introduced into the test cage with a regular food pellet (Chow), or anorexigenic agents (CCK, LiCl, LPS) allowed for feeding for 20 min. The weight of the food pellet, including the food debris left in the cage floor after test, was measured to calculate the food intake. For the fed feeding test, mice were not food deprived before testing and allowed to feed for 20 min. For optogenetic experiments, the light was started just before the mice were introduced into the testing cage or home cage. The feeding behavior was videotaped and manually analyzed.

**Feeding behavior for pharmacogenetic assay**. For chemogenetic stimulation studies, mice were injected with recombination AAV expressing hM3Dq 4 weeks before experiment. The mice were then given ad libitum with Chow 1 week prior to, during, and after the behavioral test. CNO (1 mg/kg) was injected twice a day for 7 days intraperitoneally. The body weight of mice was measured every day between 1 p.m. and 5 p.m.

For chemogenetic inhibition assay, mice were injected unilaterally into the aIC with recombination AAV expressing hM4Di 4 weeks before experiment. Mice were given ad libitum access to food prior to, during, and after the assay. Cisplatin (4 mg/kg) was administered every 2 days intraperitoneally for 2 weeks. CNO (1 mg/kg) was injected intraperitoneally twice a day for 7 days after Cisplatin administered for 1 week. The body weight of mice was measured every day between 1 p.m. and 5 p.m.

**Taste sensitivity for optogenetic assay**. Taste sensitivity was measured with free-moving mice in behavioral chamber. Prior to behavior test, mice were water deprived for 24 h. Before test mice were briefly anesthetized with isoflurane and coupled with optic fibers. 20 min after recovery, mice were introduced into the test cage with water, 0.01 mM quinine, 0.1 mM quinine, and 1 mM quinine for 20-min drinking test. The total numbers of lick were videotaped and manually analyzed.

**Water consumption assay**. Mice were water-restricted for 24 h in their home cage, acclimated to the behavioral chamber for 15 min, and then provided with access to water for 20 min. The weight of the water, total number of licks, and percent of drinking time after test was measured to calculate.

**Elevated plus-maze test**. The elevated plus-maze consisted of a plus-shaped platform with four intersecting arms: two opposing open arms and two closed arms. Animals were placed in the center of the apparatus facing a closed arm and allowed to freely explore the maze for 5 min. The parameters such as time and distance traveled in the open arm were analyzed with the smart3.0 (Panlab) software. The area was cleaned with 75% ethanol between tests.

**Open field test**. OFT was performed in an open field arena (50 cm long, 50 cm wide, and 60 cm high). Experiments were conducted under low light conditions in order to minimize anxiety effects. Mice were allowed to freely explore for 5 min.

The parameter center entries time was analyzed with the smart 3.0 software. The area was cleaned with 75% ethanol between tests.

**Real-time place preference assay**. Place preference training was performed in a custom-made two-compartment conditioned place preference (CPP) apparatus (30 cm × 25 cm × 20 cm). After connecting with optical fiber, mice infected with AAV-hChR2 or AAV-eGFP were placed in the CPP training apparatus for 20 min to assess their baseline place preference. During the test, we assigned the counterbalanced side of the chamber as the stimulation side, and placed the mice in the non-stimulated side to start the experiment. When the mouse crossed to the stimulated side of the chamber, it triggered 5 Hz laser stimulation (473-nm, 10 ms pulses) until the mouse crossed back to the non-stimulated side. Avoidance score was calculated by subtracting the time spent in stimulation side during baseline (without light) from the time spent in stimulation side during the test (with light).

**Measurement of heart rate**. Mice were anesthetized with 5% isoflurane (vol per vol) in oxygen and then kept with 1.5% isoflurane. Heart rate data were analyzed after band pass filtering (10–200 Hz, Butterworth three-order filter). Heart beats per minute (BPM) were determined by custom-written Matlab program[70,71]. At least, three calculations from each animal were averaged over a period of 3 s recording or longer.

**Data analysis**. For photometry assay, all data were analyzed with custom-written Matlab program. For photometry data, all responses were normalized to baseline using the function: $\Delta F/F = (F - F_0)/F_0$, in which $F_0$ is the median fluorescence of the baseline. The baseline period for full experiments was 30 min before time zero. To analyze the responses during feeding, the control time window was set 2 s before food touch onset to minimize potential false-positive effects. $\Delta F/F$ values are presented as average plots with a gray area indicating the ±SEM. Locomotor activity for mouse was analyzed using a video tracking software (Smart v3.0). We then averaged the value > 10 trials for food consumption. To analyze the responses during aversive stimulation, the control time window was set 60 s before IP injection and averaged the value for 3–4 mouse for aversive stimulation test.

For rabies tracing quantification, the rabies-labeled inputs into brain regions were quantified and normalized to the sum of starter cells in LH. Three brains were used for each experiment.

For axon projection quantification, images were obtained and quantified with three junction sections for each region and were averaged to obtain the final value. Three brains were used for each experiment.

Quantification of Fos staining was performed on every 6th slice in the LH from Bregma −0.7 to −1.92 mm (six sections per mice) and on every third slice in the following area: PBN from Bregma −5.02 to −5.40 mm (three sections per mice), NTS from Bregma −6.96 to −7.48 mm (five sections per mice), BLA from Bregma −0.58 to −1.34 mm (six sections per mice), mPFC from Bregma +1.98 to +1.54 mm (five sections per mice), and VPMpc from Bregma −1.82 to −1.94 mm (two sections per mice). Quantification of Fos staining in the aIC in Fig. 1a–d and Supplementary Fig. 3d, or quantification of colocalization of Fos with CamKII or GAD67 in Fig. 1e, f was performed on every 6th slice in the aIC from Bregma +1.18 to +0.02 mm (four sections per mice). Quantification of Fos staining in the whole IC in Supplementary Fig. 2a–f was performed on every third slice from Bregma +2.46 to −1.06 mm (28 sections per mice). Quantification of Fos staining in the aIC in Supplementary Fig. 2g, h was performed on every third slice in the IC from Bregma +1.18 to +0.5 mm (seven sections per mice). All images were subsequently overlaid with the corresponding atlas section to anatomically define the regions of interest. Positive cells lying on the boundary were excluded. A cell was considered positive only if it displayed an intensity value above the intensity threshold of the background. Six brains were used for each experiment. Quantification was performed using the cell counter tool in ImageJ.

**Statistical analysis**. Statistical analyses were performed using Matlab or Prism 7 (GraphPad). Throughout the paper, the level of significance is indicated as *$P$ < 0.05, **$P$ < 0.01, ***$P$ < 0.005. No statistical methods were used to predetermine the sample sizes. Pairwise comparisons were calculated with unpaired two-tailed $t$ tests, and multiple group data comparisons were calculated with one-way or two-way ANOVA with Bonferroni post hoc test. Normality was assessed with Shapiro–Wilk tests. If normality tests failed, Mann–Whitney or Wilcoxon rank-sum tests were used. Cell count, anatomical, and behavioral analysis were fully and/or partially conducted by experimenters blind to experimental conditions.

**Reporting summary**. Further information on research design is available in the Nature Research Reporting Summary linked to this article.

## Data availability
The source data underlying Figs. 1b, d, f, 2e, g, h, k, 3c, e, g, 4c, e, 5h, i, 6b, 7c, e, g, j, l, and n and Supplementary Figs. 1b, 2b–f, h, 3b–d, 4c, e, 5b, d, f, i, k, m, 6b, d, 7b, c, e, g, 8b, c, e, g, 9c, 10b, d, 11b, c, 12c, e, f and 13b are provided as a Source Data file. All relevant data are available from the corresponding authors upon reasonable request.

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

## Acknowledgements

This work was supported by the National Natural Science Foundation of China (91732102, 81471125, and 81671049 to S.Q., 31900722 to Y.W.), the Zhejiang Science Fund for Distinguished Young Scholars (LR16C090001 to S.Q.), the Fundamental Research Funds for the Central Universities of China (2019XZZX001-01-14 to S.Q.), the Chinese Ministry of Education Project 111 Program (B13026 to S.Q.), and Certificate of China Postdoctoral Science Foundation Grant (2018M630665 to Y.W.). CIHR operating grant (MOP-124807) and project grant (PJT-148648), Azrieli Neurodevelopmental Research Program and Brain Canada (M.Z.). We thank Dr. Shumin Duan (Zhejiang University) for providing us with *Slc17a6-ires-Cre* mice and Dr. Jianhong Luo (Zhejiang University) for providing us with *Camk2a-Cre* mice. We thank Yudong Zhou (Zhejiang University), Hailan Hu (Zhejiang University), and Tatiana Korotkova (Max Planck Institute for Metabolism Research) for valuable discussion. We also thank the technical support by the Core Facilities, Zhejiang University School of Medicine.

## Author contributions

Y.W., C.W.C., and M.C. performed the viral injection, animal behavior, and immunostaining experiments; collected, analyzed, and interpreted data; and participated in writing the paper. X.Y.L. performed the electrophysiological experiments; collected, analyzed, and interpreted data; and participated in writing the paper. K.Q. and H.T.W. performed immunohistochemical staining. Y.W. and L.F.J. performed the calcium-imaging experiments; and collected, analyzed, and interpreted data. L.Y. and M.Z. contributed intellectually and revised the manuscript. S.Q. was responsible for the overall supervision of the study; designed the experiments; analyzed and interpreted data; and revised the paper.

## Competing interests

The authors declare no competing interests.
