## [Peer Review File · Nature Communications]

Reviewers' comments:

Reviewer #1 (Remarks to the Author):

Unilateral Contribution of the anterior insula to feeding responses to aversive visceral stimuli in adult mice

Yu Wu, Changwan Chen, Ming Chen, Xinyou Lv, Kai Qian, Haiting Wang, Lifei Jiang, Lina Yu, Min Zhuo, and Shuang Qiu

SUMMARY:

The paper delivers a potentially important message regarding the role of excitatory populations of the insula in the regulation of feeding in pathophysiological conditions, such as in anorexia nervosa or cachexia in patients undergoing cancer treatment. Importantly, this study is the first to provide evidence that the caudal segment of the anterior insula in mice sends excitatory projections to the lateral hypothalamus, that regulate the attribution of negative valence to feeding and place preference behaviors.

The authors provide immunohistological evidence in mice, that c-fos activation in the caudal segment of the right, but not the left anterior IC, increases in response to the administration of drugs or compounds that result in the induction of malaise, such as LiCl, LPS and cisplatin. Interventions resulting in behavioral (feeding/fasting/re-feeding) or pharmacological (CCK, Ghrelin) manipulation of satiety levels, were not found to induce significant induction activity or lateralization in this segment of the insula. It is important to note that many experiments that uses LiCl as UCS in CTA did not report on such strong phenotype using similar methods. In separate experiments where in vivo fiber optometry was utilized to monitor activity in excitatory (CaMKII+) neurons of the region using GCaMP6f, the lateralization of the response to LiCl and cisplatin was further demonstrated. Even though the aforementioned compounds were also found to differentially modulate c-fos activity in a number of other cortical and sub-cortical regions, the authors did not observe any evidence of lateralization in these responses.

Results in the aforementioned correlative studies, prompted the authors to examine whether activity in excitatory neurons in this segment of the IC is sufficient to modulate feeding and place preference behaviors. In vivo optogenetic activation of CaMKII neurons of the right aIC in fasted mice resulted in the suppression of feeding behavior, and increased the expression of escape behavior towards the photo-stimulated chamber. Conversely, inhibition of this subset of excitatory neurons, promoted feeding in fed mice and increased the expression of preference to the photo-inhibited chamber. These effects were further shown to be unique to the right IC, as the aforementioned interventions were of no consequence in treated animals when applied to the left IC.

Aberrant activity and connectivity in hypothalamic and IC circuits in humans, has been implicated in pathophysiological conditions associated with suppression of feeding and the dysregulated attribution of aversive valence towards neutral stimuli. The authors thus attempted to examine whether glutamatergic neurons projecting from this caudal segment of the aIC to the lateral hypothalamus are causally involved in the attribution of negative valence to feeding and place preference behaviors. Studies were conducted in which the expression of optogenetic tools was confined within the right IC-to-LH circuit, using Cre-dependent retro-virus based constructs in vGluT2-Cre mice. These latter behavioral studies using optogenetic stimulation and inhibition, largely mirrored the earlier results using CaMKII-Cre mice, underlying the sufficiency of the circuit in mediating these effects. Data presented in the main and supplementary figures, indicate the circuit to be specifically involved in the regulation of feeding and place aversion/escape, and not extend to drinking and taste, mating or anxiety behaviors (Supplementary Figures 1-4).

The results are novel and of interest for wider field. However, the study needs further experiments and analysis to strengthen the conclusions.

MAJOR COMMENTS:

- 1) In general, there is no information regarding the specific statistical tests employed, neither in the results or the figure legends, even though the authors do indicate that t-tests and ANOVAs were used in the Methods section. It is generally expected that F and d values are reported for specific ANOVAs and whether one or two-way.
- 2) The authors do not explain why even though they injected at Bregma 0.5 (according to methods), they only displayed images from Bregma 0.1 throughout the Figures.
- 3) Did the authors well control for where the IP injection in the mice body? If the experimenters

hold the mice in left hand and inject with right it may explain the results regardless of the controls, (only what cause pain would activate one side unilaterally)?

4) No data but quantification is given from the photometry experiments (i.e. representative of a picture?).

5) Based on the current literature, it is unclear to me whether the role of this circuit does not extend to physiological conditions – do we know that hypo-activation of this circuit is strictly physiological? Given results with Ghrelin and CCK treatments, where a relatively low dose was utilized, I would suggest that the insula might in fact be involved in regulating natural feeding behaviors, but perhaps not in a lateralized manner (Figures 1 and 5).

6) The relation to human diseases: the insula is hyper-activated in response to food images in anorexic and bulimic patients. It is not hyper-activated in responses to feeding or tastant drinking itself. 1. This could be an important caveat considering the results and conclusions of this study. Results from other basic 2, as well as clinical studies 5, are not consistent with the anterior insula being limited to negative valence attribution and thus food intake inhibition. A number of studies demonstrated transcription at the anterior insula to be induced by novel appetitive stimuli in a lateralized fashion and cell-type specific fashion 3, which would suggest that (a) lateralization might be a common mechanism employed by distinct segments of the insula to encode salience, novelty and valence (b) distinct rostro-caudal segments of the anterior insula respond differently to different core characteristics of stimuli (e.g. valence 4). It is possible that the authors managed to effectively target the “visceral hot spot” of the insula, reported elsewhere in studies using rats 5,6 . It would thus be particularly important for the current state of the field that the precise characteristics of the responses are precisely mapped out in relation to the neuroanatomy of the region (this is not the case in the current version of the manuscript).

7) Injection sites provided indicate that the authors did indeed target the insula, however a major caveat is the fact that slices displayed are taken from about 0.4mm caudally, which, is a substantial distance considering the size of the murine brain. On the other hand, it diagrammatic representations of the insula suggest that a much more rostral segment was targeted. In addition, even though representative images are displayed from slices at Bregma 0.1, there is no indication as to which 3 slices are used for quantification between Bregma 0.5 and 0.1, which is particularly important considering slice thickness is 40microns. Finally, given the relative subtlety of effects even with chronic treatments, such as chemogenetic inhibition or excitation, supplementary images from injection sites indicate that at least part of the effect is not specific to the insula (see Supplementary Figure 5).

MINOR COMMENTS:

Figure 1:

- Indicate Bregma for (a) and (c).
- Results in (b) indicate the aIC to respond to Ghrelin and CCK treatments, but not in a lateralized manner.
- Scale in (d) is not appropriate for any meaningful comparison among groups. When zooming in, one can observe a pattern whereby Home Cage=Re-Fed, and tend to differ from Fasted mice.
- Indicate Bregma for (e) and (f). In the same sub-figures, it is unclear whether the graph comes from quantification of the images displayed or over 3 slices as described in Methods. It would also be useful to show what is going on within distinct sub-regions and layers in the images displayed.

Figure 4:

- Provide injection sites for (a).
- Good work on (b) and (c) given that result in (b) was weak.
- Provide injection sites for (d).
- The use of 0.5ug/kg of CCK is justified through this experiment, but is a weak dose according to literature.

Figure 5: The authors do not provide evidence of quantification of c-fos. It is unclear from the legends and the results sections, whether the effect is uniformly observed across Bregma -0.7 to -1.94.

1 - Holsen et al. 2012 <http://jpn.ca/vol37-issue5/37-5-322/>

2 - Livneh et al. 2017 <https://www.ncbi.nlm.nih.gov/pmc/articles/PMC5577930/>

3 - Inberg et al. 2016 <https://www.ncbi.nlm.nih.gov/pmc/articles/PMC4746785/>

4 - Adaikkan and Rosenblum 2015 <https://www.ncbi.nlm.nih.gov/pmc/articles/PMC4703067/>

5 - Frank et al 2017 <https://www.ncbi.nlm.nih.gov/pmc/articles/PMC5314116/>

6- Blonde et al 2015 <https://journals.plos.org/plosone/article?id=10.1371/journal.pone.0117515>

Reviewer #2 (Remarks to the Author):

In the manuscript, Wu et al. wanted to investigate the role of cortical circuits, specifically the anterior insular cortex (aIC), in mediating alterations in feeding behavior to neurological threats. The authors show that the right-side aIC (CaMKII+) neurons were activated by aversive stimuli and not satiety signals. Activation of these neurons led to suppression of food intake, and this could be reversed through inhibition of this same neuronal population. Left-side aIC manipulations had no effect on feeding behavior. The authors also show that right hemisphere CaMKII+ aIC neurons project to vGluT2+ neurons in the lateral hypothalamus, and it is this pathway that controls feeding suppression. The authors uncovered an unknown circuit that bypasses the energy homeostasis system to attenuate food intake in pathological conditions. The work presented in this study is well-controlled and conclusions made were strongly supported by the data presented. The authors elucidated an otherwise unknown circuit from the aIC to the LH that controls feeding behavior in response to aversive stimuli. Therefore, I support this manuscript for publication, but the authors need to address some concerns.

1. Comparing between mice expressing hChR2 and eArch3.0 in terms of the raster plots showing feeding processes by one animal in its home cage (Both Figure 2F and O; Figure 6 D and K) feeding time under control light (top) is shorter when eArch3.0 is expressed compared to hChR2. This can be seen in both Figures 2 and 6. Is this something biological and a consequence of the type of opsin being expressed? Furthermore, are the animals age matched for these behavioral tests? The authors need to clarify this point.

2. Figure 4: CNO injection inhibited (rapidly) food intake in fasted mice expressing hM3Dq in the right aIC neurons during a 1-hr test. Is this a time dependent effect? Were other times tested? The authors should comment (or provide more time points) illustrating just how "rapid" food intake is inhibited.

3. Figure 4: a saline injection control (or more preferably a CNO injection control in non DREADD expressing mice) for food intake behavioral experiments needs to be included.

4. Figure 5: Panels C and D. The excitatory current in Panel C and Panel D is the exact same trace. If these experiments were performed in the same neuron (optogenetics and +CNQX) this needs to be clarified. In addition the color scheme is confusing, Panel D should be a different color than red (it confuses the reader due to the fact that red in Panel C denotes HP@0mV)

Reviewer #3 (Remarks to the Author):

This is a strong paper demonstrating that right (but not left) anterior insular cortex (raIC) neurons respond to aversive signals and inhibit food intake by axon projections to glutamatergic neurons in the lateral hypothalamus. The authors examine both short and long-term effects of inhibiting or activating the raIC neurons. The laterality of the IC is interesting in its own right. The fact that they connect the raIC neurons to glutamatergic LH neurons that have already been shown to inhibit food intake when activated is another strength of this paper.

1. The authors study the projections of the raIC neurons but they fail to examine the relevant inputs to these, except to say in the introduction that IC receives input from the thalamus. It would be useful to know more precisely what thalamic neurons project to the raIC neurons examined in this paper.

2. The parabrachial CGRP-expressing neurons are activated by all of the threats mentioned in this paper and they send projections to the VPMpc thalamus and IC; however, they also respond to the food shocks, CCK and large meals (after a fast) that do not seem to activate the raIC neurons. A little speculation on how some but not other aversive signals reach the raIC seems warranted.

3. The authors demonstrate that the raIC is preferentially activated by a variety of aversive signals but they never say whether this laterality extends to the LH. Fig. 5A shows that the right aIC projects to right LH, but do LPS, LiCl, etc selectively activate the right LH?

Editorial comments

Line 37, In what sense is this circuit "novel?"

Lines 64-74 are not necessary. They only rehash to Abstract

Line 79, There is only one Fos gene in the mouse, therefore, no need to distinguish c-Fos from viral or v-Fos.

Line 83, a Ref to Cisplatin results is needed

Line 90, consider saying "either the right or the left aIC"

Line 91 (and lots of other places), 24-hr fasted mice. Insert a hyphen when using nouns as adjectives. See also lines 116, 137, 138

Line 101 CamKII-positive neurons

Line 112, What is meant by AAV2/8? The numbers refer to serotypes. The serotype is probably either 2 or 8, but not both

Line 113, Only approved mouse gene names should be used in italics. All mouse gene names start with a capital letter followed by lowercase letters and numbers. CamKII should be Camk2a (as in Methods). I am not sure which gene was targeted for these mice. Alternatively, the authors can use the style they have but without italics.

Line 122-125. This is an awkward sentence. The sympathetic nervous system is not is not a classical consummatory nucleus as implied by this sentence.

Line 140, What is meant by "feeding tendency"? It either did or did not enhance feeding. See also line 216.

Line 168-169, Consider, "Body weight returned to normal after cessation of CNO treatment"

Line 173, Consider "Prevented LiCl and LPS induced reduction of food intake"

Line 188, Central should not be capitalized

Line 208, see comment line 133. Vglut2 should be Slc17a6 (italics) or do not use italics

Line 217, "while (it) had no effects..."

Line 241, "hijacks" is not the best word here. Perhaps "interrupts" is meant

Lines 255 & 300, Ref style

Line 883, cage, not "cages"

Line 885, What does "times activation to stop feeding" mean?

Fig. 2n, p, 3c, 4b, 6c, e, j, l Put (%) and the end of ordinate label

Fig. 2p, 6l, what does "Times activation to feeding" mean?

Fig. 5 legend, some abbreviations used in the figure are not defined.

Reviewed by R. Palmiter

Reviewers' comments:

Reviewer #1 (Remarks to the Author):

Unilateral Contribution of the anterior insula to feeding responses to aversive visceral stimuli in adult mice

Yu Wu, Changwan Chen, Ming Chen, Xinyou Lv, Kai Qian, Haiting Wang, Lifei Jiang, Lina Yu, Min Zhuo, and Shuang Qiu

SUMMARY:

The paper delivers a potentially important message regarding the role of excitatory populations of the insula in the regulation of feeding in pathophysiological conditions, such as in anorexia nervosa or cachexia in patients undergoing cancer treatment. Importantly, this study is the first to provide evidence that the caudal segment of the anterior insula in mice sends excitatory projections to the lateral hypothalamus, that regulate the attribution of negative valence to feeding and place preference behaviors.

The authors provide immunohistological evidence in mice, that c-fos activation in the caudal segment of the right, but not the left anterior IC, increases in response to the administration of drugs or compounds that result in the induction of malaise, such as LiCl, LPS and cisplatin. Interventions resulting in behavioral (feeding/fasting/re-feeding) or pharmacological (CCK, Ghrelin) manipulation of satiety levels, were not found to induce significant induction activity or lateralization in this segment of the insula. It is important to note that many experiments that uses LiCl as UCS in CTA did not report on such strong phenotype using similar methods. In separate experiments where in vivo fiber optometry was utilized to monitor activity in excitatory (CaMKII+) neurons of the region using GCaMP6f, the lateralization of the response to LiCl and cisplatin was further demonstrated. Even though the aforementioned compounds were also found to differentially modulate c-fos activity in a number of other cortical and sub-cortical regions, the authors did not observe any evidence of lateralization in these responses.

Results in the aforementioned correlative studies, prompted the authors to examine whether activity in excitatory neurons in this segment of the IC is sufficient to modulate feeding and place preference behaviors. In vivo optogenetic activation of CaMKII neurons of the right aIC in fasted mice resulted in the suppression of feeding behavior, and increased the expression of escape behavior towards the photo-stimulated chamber. Conversely, inhibition of this subset of excitatory neurons, promoted feeding in fed mice and increased the expression of preference to the photo-inhibited chamber. These effects were further shown to be unique to the right IC, as the aforementioned interventions were of no consequence in treated animals when applied to the left IC.

Aberrant activity and connectivity in hypothalamic and IC circuits in humans, has been implicated in pathophysiological conditions associated with suppression of feeding and the

dysregulated attribution of aversive valence towards neutral stimuli. The authors thus attempted to examine whether glutamatergic neurons projecting from this caudal segment of the aIC to the lateral hypothalamus are causally involved in the attribution of negative valence to feeding and place preference behaviors. Studies were conducted in which the expression of optogenetic tools was confined within the right IC-to-LH circuit, using Cre-dependent retro-virus based constructs in vGluT2-Cre mice. These latter behavioral studies using optogenetic stimulation and inhibition, largely mirrored the earlier results using CaMKII-Cre mice, underlying the sufficiency of the circuit in mediating these effects. Data presented in the main and supplementary figures, indicate the circuit to be specifically involved in the regulation of feeding and place aversion/escape, and not extend to drinking and taste, mating or anxiety behaviors (Supplementary Figures 1-4). The results are novel and of interest for wider field. However, the study needs further experiments and analysis to strengthen the conclusions.

We sincerely appreciate the reviewer's elaborate summary and valuable comments. The following suggestions help us to open our mind and think more carefully about our work. More importantly, these comments will guide us in our future work.

MAJOR COMMENTS:

1) In general, there is no information regarding the specific statistical tests employed, neither in the results or the figure legends, even though the authors do indicate that t-tests and ANOVAs were used in the Methods section. It is generally expected that F and d values are reported for specific ANOVAs and whether one or two-way.

Response

Thanks to these comments. In the revised manuscript, we have included both the statistical tests employed and the related *F* and *p* values in Figure Legends.

2) The authors do not explain why even though they injected at Bregma 0.5 (according to methods), they only displayed images from Bregma 0.1 throughout the Figures.

Response

Sorry for this misleading information. In our experiment, based on the injection site and injection volume, the virus was approximately diffused to the area from Bregma+1.54 to Bregma-0.10. In the first version of manuscript, we just chose one of images (such as from Bregma+0.1) to indicate successful infection of the virus. In the revised manuscript, we have replaced these images with those from the injection site (Bregma+0.5, AP) to avoid obscurity.

3) Did the authors well control for where the IP injection in the mice body? If the experimenters hold the mice in left hand and inject with right it may explain the results regardless of the controls, (only what cause pain would activate one side unilaterally)?

Response

We agree with the reviewer's concern and performed experiments to examine whether IP injection site determines the unilateral activation of the aIC. As shown in Supplementary Fig. 1g, the right aIC was significantly activated no matter whether LiCl was injected in the left side or right side of the mice body, suggesting that unilateral activation of the aIC is not due to the site of IP injection.

4) No data but quantification is given from the photometry experiments (i.e. representative of a picture?)

Response

Thanks for this comment. In the revised manuscript, we have included the representative data and the related statistical analysis from the photometry experiments in Supplementary Fig. 2b-e.

5) Based on the current literature, it is unclear to me whether the role of this circuit does not extend to physiological conditions – do we know that hypo-activation of this circuit is strictly physiological? Given results with Ghrelin and CCK treatments, where a relatively low dose was utilized, I would suggest that the insula might in fact be involved in regulating natural feeding behaviors, but perhaps not in a lateralized manner (Figures 1 and 5).

Response

We appreciate the reviewer's comment.

Previously, we utilized Ghrelin (0.4 mg/kg) and CCK (0.5 µg/kg) to pharmacologically manipulate satiety levels. Ghrelin (0.1~1.0 mg/kg) has been reported to increase food intake in rat^{1, 2}, Siberian Hamsters³, and mice^{4, 5}, and CCK (0.1~0.5 µg/kg) has been reported to inhibit food intake in rat and mice^{6, 7} and in Siberian sturgeon⁸. Here, we did additional experiments using relatively higher dose of Ghrelin (1.0 mg/kg) and CCK (5 µg/kg). We did observe the activation of aIC under these treatment, but in a much less degree as compared to LiCl or LPS treatment (Fig. 1a-b and Supplementary Fig. 1c in the revised manuscript). Moreover, LiCl and LPS treatment produced a robust induction of Fos expression in the right aIC within a narrow segment (between 1.18 and 0.02 mm anterior of Bregma), whereas CCK and Ghrelin treatment resulted in a sparse and distributed pattern of Fos expression along both sides of the aIC (Fig. 1a, b and Supplementary Fig. 1c, d). We have replaced previous data (Ghrelin, 0.4 mg/kg and CCK, 0.5 µg/kg) with these new data (Ghrelin, 1.0 mg/kg and CCK, 5 µg/kg).

We did not observe obvious Fos⁺ expression in the aIC under natural condition (feeding/fasted/refeeding). However, it should be noted that Fos staining may not detect all the neurons that are activated. Our previous results after activation or silencing of the right aIC^{CamKII} neurons suggest that these neurons can not only mediate aversive visceral stimuli-associated anorexia but also bidirectionally modulate feeding behaviors in general. Therefore, we agree with the reviewer that the aIC may be involved in regulating natural feeding behaviors in some way.

We have modified the related abstract, introduction, and results, and added comments in the discussion section. We avoid using “physiological conditions” or “pathological conditions” to restrain the role of aIC in regulating feeding behaviors.

6) The relation to human diseases: the insula is hyper-activated in response to food images in anorexic and bulimic patients. It is not hyper-activated in responses to feeding or tastant drinking itself. 1. This could be an important caveat considering the results and conclusions of this study. Results from other basic 2, as well as clinical studies 5, are not consistent with the anterior insula being limited to negative valence attribution and thus food intake inhibition. A number of studies demonstrated transcription at the anterior insula to be induced by novel appetitive stimuli in a lateralized fashion and cell-type

specific fashion 3, which would suggest that (a) lateralization might be a common mechanism employed by distinct segments of the insula to encode salience, novelty and valence (b) distinct rostro-caudal segments of the anterior insula respond differently to different core characteristics of stimuli (e.g. valence 4). It is possible that the authors managed to effectively target the “visceral hot spot” of the insula, reported elsewhere in studies using rats 5,6 . It would thus be particularly important for the current state of the field that the precise characteristics of the responses are precisely mapped out in relation to the neuroanatomy of the region (this is not the case in the current version of the manuscript).

Response

We sincerely appreciate the reviewer’s comments.

Here, to examine whether there is some “visceral hot spot” in the mouse insular cortex, we systematically analyzed the distribution of Fos⁺ neurons in different sub-regions of the insular cortex in response to different stimuli. Interestingly, aversive visceral stimuli, such as LiCl and LPS, significantly activated neurons within a narrow segment (caudal part) of the right aIC (Bregma+1.18 to Bregma+0.02) (Supplementary Fig. 1c-d).

We usually injected virus at site (+0.5 mm antero-posterior (AP); ±3.85 mm medio-lateral (ML); -2.82 mm dorso-ventral (DV) relative to bregma) and implanted a fiberoptic cannula above the injected site. Therefore, we have already targeted this caudal segment of the aIC and demonstrated that this caudal segment of the aIC is involved in regulating food consumption. To further confirm the specificity of this segment in regulating feeding behavior, we injected AAV8-DIO-hChR2-eGFP into other two sites neighbored to this caudal segment of the aIC. One is the right posterior insular cortex (pIC) (-1.10 mm AP; -3.90 mm ML; -3.25 mm DV relative to bregma), the other is the rostral segment of the right aIC (+1.6 mm AP; -3.1mm ML; -1.8 mm DV relative to bregma). As shown in Supplementary Fig. 4, photoactivation of the right pIC or the rostral segment of the right aIC has no effect on food consumption in 24-h fasted mice. Combined together, these results indicate that the caudal segment of the right aIC acts as “hot spot” for aversive visceral stimuli. This “visceral hot spot” is just located between the “bitter center” and the “sweet center” identified by Zuker’s group⁹, further indicating the functional heterogeneity of the insular cortex.

We have cited the related references in the revised manuscript and added some comments related to these findings in the Discussion Section (p12).

7) Injection sites provided indicate that the authors did indeed target the insula, however a major caveat is the fact that slices displayed are taken from about 0.4mm caudally, which, is a substantial distance considering the size of the murine brain. On the other hand, it diagrammatic representations of the insula suggest that a much more rostral segment was targeted. In addition, even though representative images are displayed from slices at Bregma 0.1, there is no indication as to which 3 slices are used for quantification between Bregma 0.5 and 0.1, which is particularly important considering slice thickness is 40microns. Finally, given the relative subtlety of effects even with chronic treatments, such as chemogenetic inhibition or excitation, supplementary images from injection sites indicate that at least part of the effect is not specific to the insula (see Supplementary Figure 5).

Response

We appreciate the reviewer's comments.

As we have explained in response to major comment 2, the virus (for optogenetics) was diffused to the area from Bregma+1.52 to Bregma-0.1, which covers the “visceral hot spot” (Bregma from +1.18 mm to +0.02 mm) in response to aversive stimuli, and the fibreoptic cannula was implanted above the injected site (+0.5 mm AP). In the first version of manuscript, we just chose one of images (such as from Bregma+0.1) to indicate successful infection of the virus. In the revised manuscript, we have replaced these images with those from Bregma+0.5 to avoid obscurity. We also double checked all of the previous data and rejected those with inappropriate expression segment or implantation sites.

We are sorry for not describing clearly the method for statistical analysis of IHC in our first version of manuscript. Here, we have included the following description as “...*Quantification of Fos staining was performed on every 6th slice in the following areas: in the IC from bregma +2.46 to -1.06 mm (14 sections per mice), LH from bregma -0.7 to -1.92 mm (6 sections per mice) and on every third slice in the following area: PBN from bregma -5.02 to -5.40 mm (3 sections per mice), NTS from bregma -6.96 to -7.48 mm (5 sections per mice), BLA from bregma -0.58 to -1.34 mm (6 sections per mice), mPFC from bregma +1.98 to +1.54 mm (5 sections per mice), and VPMpc from bregma -1.82 to -1.94 mm (2 sections per mice). Quantification of colocalization of Fos with CamKII or GAD67 was performed on every 6th*

slice in the aIC from bregma +1.18 to +0.02 mm (4 sections per mice)...” in the Methods section (p23).

For DREADD experiments, we agree with the reviewer’s concern and have double checked all of the data. We have included new data and chose those with appropriate virus distribution in the caudal segment of the aIC (mainly from Bregma+1.34 to Bregma+0.02). The final results were shown in Fig. 4c, 4g and Supplementary Fig. 7, and the conclusions are consistent with the previous ones.

MINOR COMMENTS:

Figure 1:

- ***Indicate Bregma for (a) and (c).***

Response

We have added Bregma for a and c in Figure 1.

- ***Results in (b) indicate the aIC to respond to Ghrelin and CCK treatments, but not in a lateralized manner.***

Response

As we have explained in response to major comment 5, we used Ghrelin (1 mg/kg) and CCK (5 µg/kg) to repeat the corresponding experiments and observed that the aIC responds to Ghrelin (1 mg/kg) and CCK (5 µg/kg) as well, but in a much less degree and in a non-lateralized manner (Fig. 1a and Supplementary Fig. 1c, d). We have replaced the previous data with these new data and modified the description in the revised manuscript.

- ***Scale in (d) is not appropriate for any meaningful comparison among groups. When zooming in, one can observe a pattern whereby Home Cage=Re-Fed, and tend to differ from Fasted mice.***

Response

We have modified the Ordinate axis of panel d in the revised manuscript. Since the number of Fos⁺ neurons is quite low, it is hard to tell whether these three groups are different from each other. It should be noted that this result is based on Fos expression, which may not detect all the neurons that are activated.

• Indicate Bregma for (e) and (f). In the same sub-figures, it is unclear whether the graph comes from quantification of the images displayed or over 3 slices as described in Methods. It would also be useful to show what is going on within distinct sub-regions and layers in the images displayed.

Response

We have indicated Bregma for these two panels (e and h in the revised manuscript) and described in detail the analysis of Fos⁺ neurons in Methods section (p23). We have also included the images of the left and right aIC from Bregma+0.26 mm and inserted the enlarged images to show the colocalization of Fos with CamKII (Fig. 1e)

Figure 4:

• Provide injection sites for (a).

Response

We have provided injection sites for (a) as panel a in the Fig. 4.

• Good work on (b) and (c) given that result in (b) was weak.

Response

Thanks to the comment.

• Provide injection sites for (d).

Response

We have provided injection sites for (d) as panel e in the Fig. 4.

• The use of 0.5ug/kg of CCK is justified through this experiment, but is a weak dose according to literature.

Response

Thanks for the comment. We have repeated this experiment with a higher concentration of CCK (5 µg/kg). As shown in Fig. 4g in the revised manuscript, inhibition of the right or left aIC^{CamKII} neurons had no effect on CCK-induced reduction of food intake in fasted mice, similar with our previous conclusion. Together with another new data that CCK (5 µg/kg)

induces activation of the aIC in a sparse and distributed pattern, it indicates that the aIC may respond differently to pharmacologically manipulated satiety when compared to aversive visceral stimuli. We have replaced the previous data with this new data in the revised manuscript.

Figure 5: The authors do not provide evidence of quantification of c-fos. It is unclear from the legends and the results sections, whether the effect is uniformly observed across Bregma -0.7 to -1.94

Response

We are sorry for it. We have described in detail the analysis of Fos⁺ neuron in Methods section as “*Quantification of Fos staining was performed on every 6th slice in the following areas:... LH from bregma -0.7 to -1.94 mm (6 sections per mice)*” (p23). We have shown images of Fos staining in different sections of LH in Fig. 5b. Here, we observed obvious Fos expression around Bregma-1.34, which is mainly due to the implantation of fibreoptic cannula (-1.28 mm AP, ±1.23 mm ML, -5.28 mm DV) above this site.

Reviewer #2 (Remarks to the Author):

In the manuscript, Wu et al. wanted to investigate the role of cortical circuits, specifically the anterior insular cortex (aIC), in mediating alterations in feeding behavior to neurological threats. The authors show that the right-side aIC (CaMKII+) neurons were activated by aversive stimuli and not satiety signals. Activation of these neurons led to suppression of food intake, and this could be reversed through inhibition of this same neuronal population. Left-side aIC manipulations had no effect on feeding behavior. The authors also show that right hemisphere CaMKII+ aIC neurons project to vGluT2+ neurons in the lateral hypothalamus, and it is this pathway that controls feeding suppression. The authors uncovered an unknown circuit that bypasses the energy homeostasis system to attenuate food intake in pathological conditions. The work presented in this study is well-controlled and conclusions made were strongly supported by the data presented. The authors elucidated an otherwise unknown circuit from the aIC to the LH that controls feeding behavior in response to aversive stimuli. Therefore, I support this manuscript for publication, but the authors need to address some concerns.

We greatly appreciate the reviewer's valuable comments.

1. Comparing between mice expressing hChR2 and eArch3.0 in terms of the raster plots showing feeding processes by one animal in its home cage (Both Figure 2F and O; Figure 6 D and K) feeding time under control light (top) is shorter when eArch3.0 is expressed compared to hChR2. This can be seen in both Figures 2 and 6. Is this something biological and a consequence of the type of opsin being expressed? Furthermore, are the animals age matched for these behavioral tests? The authors need to clarify this point.

Response

We thank the reviewer for these comments. We have carefully matched the animal age for these behavioral tests. This difference is mainly due to the status of the mice. When eArch3.0 is expressed, the mice are satiated before testing, while when hChR2 is expressed, the mice are 24-hr fasted before testing. That is why the feeding time under control light is shorter when eArch3.0 is expressed compared to hChR2. We have clarified it in the revised manuscript.

2. Figure 4: CNO injection inhibited (rapidly) food intake in fasted mice expressing hM3Dq in the right aIC neurons during a 1-hr test. Is this a time dependent effect? Were other times tested? The authors should comment (or provide more time points) illustrating just how “rapid” food intake is inhibited.

Response

Thanks for the suggestion.

We have repeated this experiment and added two more time points (20 min and 40 min). We observed that food intake was already reduced during the first 20 min after CNO injection in the 24-h fasted mice expressing hM3Dq in the right aIC neurons. It indicates that CNO injection rapidly inhibits food intake. These data are shown in Fig. 4c in the revised manuscript.

3. Figure 4: a saline injection control (or more preferably a CNO injection control in non DREADD expressing mice) for food intake behavioral experiments needs to be included.

Response

Thanks for the suggestion.

We have included a saline injection control in the revised manuscript. As shown in Supplementary Fig. 7c, saline injection had no effect on feeding behaviors.

4. Figure 5: Panels C and D. The excitatory current in Panel C and Panel D is the exact same trace. If these experiments were performed in the same neuron (optogenetics and +CNQX) this needs to be clarified. In addition the color scheme is confusing, Panel D should be a different color than red (it confuses the reader due to the fact that red in Panel C denotes HP@0mV)

Response

Thanks for the suggestion.

In the first version of manuscript, the excitatory current in Panel C and Panel D is from the same neuron (optogenetics and +CNQX). To avoid obscurity, we have used trace from another neuron and modified the color of Panel D in the revised manuscript.

Reviewer #3 (Remarks to the Author):

This is a strong paper demonstrating that right (but not left) anterior insular cortex (raIC) neurons respond to aversive signals and inhibit food intake by axon projections to glutamatergic neurons in the lateral hypothalamus. The authors examine both short and long-term effects of inhibiting or activating the raIC neurons. The laterality of the IC is interesting in its own right. The fact that they connect the raIC neurons to glutamatergic LH neurons that have already been shown to inhibit food intake when activated is another strength of this paper.

We greatly appreciate the reviewer's valuable comments.

1. The authors study the projections of the raIC neurons but they fail to examine the relevant inputs to these, except to say in the introduction that IC receives input from the thalamus. It would be useful to know more precisely what thalamic neurons project to the raIC neurons examined in this paper.

Response

Thanks to the reviewer's suggestion. In the revised manuscript, we injected CTB555 to the right aIC and observed significant signals in the VPMpc thalamus 7 days after injection. We further demonstrated that the projecting neurons from VPMpc to aIC are primarily glutamatergic via in situ hybridization. We have added these data in the revised manuscript as Supplementary Fig. 8c-d.

2. The parabrachial CGRP-expressing neurons are activated by all of the threats mentioned in this paper and they send projections to the VPMpc thalamus and IC; however, they also respond the food shocks, CCK and large meals (after a fast) that do not seem activate the raIC neurons. A little speculation on how some but not other aversive signals reach the raIC seems warranted.

Response

Thanks to these valuable comments.

In our experiments, we also observed the bilateral activation of VPMpc thalamus and PBN nucleus in respond to pathological stimuli (such as LiCl, LPS, cisplatin, Supplementary Fig. 1a-b). However, we found that the caudal segment of the right anterior insular cortex is a “visceral hot spot” to sense the aversive visceral stimuli, but not aversive foot shocks, CCK or large meals. The insular cortex processes multisensory information, including visceral, gustatory, somatosensory, and auditory modalities. Interestingly, one recent research has identified a crucial role of posterior IC in threat learning¹⁰. Moreover, sweet and bitter taste centers have been identified to be located within different segment of the insular cortex⁹. These data demonstrate the topographic segregation in the functional architecture of the insula. It is possible that inputs carrying information from outside and inside the body project to topographically organized insular sensory regions, giving rise to the “visceral insular cortex”, “the gustatory cortex” and insular somatosensory fields. More tracing work is required to carefully analyze the connection between the IC and thalamus, PBN or other brain regions (such as the neuronal types, subregion, left-right difference, etc.) that serves sensory functions. We have included these comments in the Discussion section.

3. The authors demonstrate that the raIC is preferentially activated by a variety of aversive signals but they never say whether this laterality extends to the LH. Fig. 5A shows that the right aIC projects to right LH, but do LPS, LiCl, etc selectively activate the right LH?

Response

Thanks to these valuable comments.

We have examined the activation of LH in response to LPS and LiCl treatment and found that both the right and the left LH were activated, indicating that LH bilaterally responds to aversive stimuli (please see the Supplementary Fig. 1a-b in the revised manuscript). It should be noted that Fos expression provides no information about neuronal types and the circuitry involved. Whether the functional role of the left-side aIC-to-LH projection is identical to or different from that of the right-side aIC-to-LH projection remains to be determined.

Editorial comments

Line 37, In what sense is this circuit “novel?”

Response

Sorry for it. We have deleted this word.

Lines 64-74 are not necessary. They only rehash to Abstract

Response

We have rewritten this paragraph as “*In this study, we demonstrate that the CamKII-positive neurons in the caudal segment of the right-side, but not the left-side, aIC respond to aversive visceral stimuli and suppress feeding via projections to the vGluT2 positive neurons in the lateral hypothalamus (LH). Thus, we identify a “visceral hot spot” in the aIC that conveys aversive visceral information to the LH to regulate food consumption.*” (p3-4)

Line 79, There is only one Fos gene in the mouse, therefore, no need to distinguish c-Fos from viral of v-Fos.

Response

We have substituted “c-Fos” with “Fos” in the revised manuscript.

Line 83, a Ref to Cisplatin results is needed

Response

We have added one reference here (*Alhadeff et,al J. Neurosci., Jan 11, 2017 37(2):362–370*).

Line 90, consider saying “either the right or the left aIC”

Response

We have modified it as “either the right or the left aIC”

Line 91 (and lots of other places), 24-hr fasted mice. Insert a hyphen when using nouns as adjectives. See also lines 116, 137, 138

Response

We have corrected them accordingly.

Line 101 CamKII-positive neurons

Response

We have corrected them accordingly.

Line 112, What is meant by AAV2/8? The numbers refer to serotypes. The serotype is probably either 2 or 8, but not both

Response

2 in AAV2/8 indicates the wild type of ITR serotypes, while 8 indicates the virus serotypes. In the revised manuscript, we have used AAV8 or AAV9 instead of AAV2/8 or AAV2/9.

Line 113, Only approved mouse gene names should be used in italics. All mouse gene names start with a capital letter followed by lowercase letters and numbers. CamKII should be Camk2a (as in Methods). I am not sure which gene was targeted for these mice. Alternatively, the authors can use the style they have but without italics.

Response

For mouse gene names, we have substituted “*CamKII*” with italic “*Camk2a*” and “*vGluT2*” with italic “*Slc17a6*”, respectively.

Line 122-125. This is an awkward sentence. The sympathetic nervous system is not is not a classical consummatory nucleus as implied by this sentence.

Response

We have modified this sentence as “*We further tested whether activation of the right aIC^{CamKII} neurons affects the valence of taste, cardiac function, or other classical consummatory behaviors and observed that optogenetic activation of the right aIC^{CamKII} neurons has no effect on the mice bitter sensitivity (Supplementary Fig. 3b), heart rate (Supplementary Fig. 3c, d), drinking (Supplementary Fig. 3e, f) or mating behaviors (Supplementary Fig. 3g-i).*”

Line 140, What is meant by “feeding tendency”? It either did or did not enhance feeding. See also line 216.

Response

We have deleted “tendency” in these two sentences.

Line 168-169, Consider, “Body weight returned to normal after cessation of CNO treatment”

Response

We have replaced the previous sentence with “*Body weight returned to normal after cessation of CNO treatment*” .

Line 173, Consider “Prevented LiCl and LPS induced reduction of food intake”

Response

We have replaced “*rescued*” with “*prevented*”.

Line 188, Central should not be capitalized

Response

Sorry for it. We have replaced “*Central amygdaloid area*” with “*central amygdaloid area*”.

Line 208, see comment line 133. Vglut2 should be Slc17a6 (italics) or do not use italics

Response

We have corrected it accordingly.

Line 217, “while (it) had no effects...”

Response

We have corrected it.

Line 241, “hijacks” is not the best word here. Perhaps “interrupts” is meant

Response

We have replaced “hijacks” with “interrupts”.

Lines 255 & 300, Ref style

Response

Sorry for it. We have corrected the reference style accordingly.

Line 883, cage, not “cages”

Response

Sorry for it. We have replaced “cages” with “cage”.

Line 885, What does “times activation to stop feeding” mean?

Response

Sorry for this confusing description. In Fig. 2f, the light (473 nm) tends to inhibit feeding behavior, so we calculated the percentage of the light stimuli that inhibits feeding. To make it clear, we used “Feeding-inhibiting stimuli (%)” as the ordinate label. In addition, we added the corresponding description in the figure legend.

Fig. 2n, p, 3c, 4b, 6c, e, j, l Put (%) and the end of ordinate label

Response

Sorry for it. We have corrected these labeling accordingly.

Fig. 2p, 6l, what does “Times activation to feeding” mean?

Response

Sorry for this confusing description. In Fig. 2o and 6l, light (525 nm) tends to induce feeding behavior rapidly, so we calculate the percentage of the light stimuli that induces feeding. To make it clear, we used “feeding-inducing stimuli (%)” as the ordinate label. In addition, we added the corresponding description in the related figure legend.

Fig. 5 legend, some abbreviations used in the figure are not defined.

Response

We have defined all the abbreviations used in the Fig. 5.

We thank all the reviewers for their perceptive and constructive comments, and hope that with these revisions and additional data, the paper will be acceptable for publication in Nature Communications

Reference

1. Wren AM, *et al.* The novel hypothalamic peptide ghrelin stimulates food intake and growth hormone secretion. *Endocrinology* **141**, 4325-4328 (2000).
2. Wren AM, *et al.* Ghrelin enhances appetite and increases food intake in humans. *J Clin Endocrinol Metab* **86**, 5992 (2001).
3. Bradley SP, Pattullo LM, Patel PN, Prendergast BJ. Photoperiodic regulation of the orexigenic effects of ghrelin in Siberian hamsters. *Horm Behav* **58**, 647-652 (2010).
4. Szentirmai E. Central but not systemic administration of ghrelin induces wakefulness in mice. *PLoS One* **7**, e41172 (2012).
5. Jerlhag E. The antipsychotic aripiprazole antagonizes the ethanol- and amphetamine-induced locomotor stimulation in mice. *Alcohol* **42**, 123-127 (2008).
6. Lo CM, *et al.* Interaction of apolipoprotein AIV with cholecystokinin on the control of food

intake. *Am J Physiol Regul Integr Comp Physiol* **293**, R1490-1494 (2007).

7. Lo CC, *et al.* Intraperitoneal CCK and fourth-intraventricular Apo AIV require both peripheral and NTS CCK1R to reduce food intake in male rats. *Endocrinology* **155**, 1700-1707 (2014).
8. Zhang X, *et al.* CCK reduces the food intake mainly through CCK1R in Siberian sturgeon (*Acipenser baerii* Brandt). *Sci Rep* **7**, 12413 (2017).
9. Peng Y, Gillis-Smith S, Jin H, Trankner D, Ryba NJ, Zuker CS. Sweet and bitter taste in the brain of awake behaving animals. *Nature* **527**, 512-515 (2015).
10. Berret E, *et al.* Insular cortex processes aversive somatosensory information and is crucial for threat learning. *Science* **364**, (2019).

Reviewers' comments:

Reviewer #1 (Remarks to the Author):

Unilateral Contribution of the anterior insula to feeding responses to aversive visceral stimuli in adult mice

Revisions R2

Yu Wu, Changwan Chen, Ming Chen, Xinyou Lv, Kai Qian, Haiting Wang, Lifei Jiang, Lina Yu, Min Zhuo, and Shuang Qiu

The manuscript is improved following the reviewers comments. However, we have the below described reservations (mainly for figures 1 and 2).

For all figures:

The Journal suggests that statistical information is included in the Results not the legends.

The time point for c-fos immunohistochemistry and the RNAscope experiments is not stated.

Figure 1:

1. The authors adjusted the doses for CCK and Ghrelin, as we recommended. From these new experiments (1a), one can note that these agents do induce activation of the insula, yet not in a lateralized fashion like LiCl, LPS and Cisplatin. However, another interesting difference not addressed by the authors is that what the authors term as "physiological agents" (i.e. CCK and Ghrelin), are activating distinct part of the insula than presumably immune response-activating agents (LiCl, LPS, Cisplatin). According to the new results, regardless of lateralization, anorexigenic agents primarily activate deep layers of the Agranular subdivision of the insula, whereas Ghrelin, the only hunger-producing hormone activates almost exclusively the Granular aIC. For these reasons, as we indicated in our first communication, it would be important to define where this change is happening (1a, c).

2. The authors state that they quantified connectivity and c-fos responses using 4 slices between Bregma 1.18 and 0.00 or 14 slices between Bregma +2.46 and -1.06. This largely over-estimates the homogeneity of the response, while differences between the aIC layers and subregions have been ignored. Complementary studies (Supplementary 4) indicate that the authors may have at least partial answers as to exactly where the lateralization is of relevance.

3. The images displayed suggest that the CaMKII staining worked partially – perhaps the authors would find it easier to do this using CaMKII-Cre mice at their disposal. The methods indicate that the authors used two rabbit antibodies (c-fos and CaMKII) together, is it a typing error?

Figure 2:

1. The construct that the authors claimed to have injected to chemogenetically inhibit the projection cannot be a Chr2 expressing AAV (AAV8-DIO-hChr2(H134R)-eGFP), as mentioned in the results section?

2. Optogenetic activation of the projection does seem to suppress feeding responses quite convincingly (f). However, the control experiments in the optogenetic inhibition experiment might require better matching with the stimulated condition. The trials shown indicate that control light exposure, eventually leads to feeding, so it is perhaps best to use a different representative trial that reflects the clarity of the differences in the data analyzed.

3. There is no information regarding the number of approaches or the percentage spent feeding. It is further unclear whether mice used here were fasted or fed and what the hypotheses entailed in these experiments were.

4. The data regarding fibre optometry in Figure 2 very elegantly demonstrate the unilaterality of the response to LiCl and cisplatin. However, what is also apparent is that LiCl or cisplatin stimulation increases activity in the projection at distinct time points (10seconds vs 30minutes). Given this, it is further highlighted that control stimulation should be more closely matched to the timing of the actual stimulation be useful.

Figure 3-6: OK.

Discussion:

1. We do not agree with the over-interpretation regarding unphysiological and physiological stimuli. Given the evidence provided by the authors themselves, the insula shows clear responses to both ghrelin and CCK (compared to saline), both physiological stimuli. Previous studies in the cytoarchitecture of the insula in fact suggest that visceral information should primarily activate the granular insula, which can be seen in the ghrelin responses. Conversely, a number of other studies

suggest the dysgranular/agranular subregions to be dedicated to emotional processing, as well as CS-US associations.

2. The aspect of local connectivity at the aIC was not addressed sufficiently, while calling the circuit a "visceral hot spot" might be problematic since the study did not use a method to cover the IC continuously.

Reviewer #2 (Remarks to the Author):

The authors had sufficiently addressed my concerns over the previous version of this manuscript. I therefore support the publication of this study.

Reviewer #3 (Remarks to the Author):

The authors have attended to most of my concerns. I have provided some line-by-line editorial suggestions below. The only new information requested deals with Fos induction in CeA in response to LiCl, LPS and Cis-platin. The authors probably have that information since the quantified Fos in the adjacent BLA.

Line 44, Consider revising your sentence: "By now, it is comparatively elusive as to the neural circuits involved in mediating feeding responses under these non-homeostatic conditions" to the following: "However, the neural circuits involved in mediating feeding responses under these non-homeostatic conditions are comparatively elusive"

Line 80, The authors show robust activation of Fos in the PBN (~ 400 Fos+ cell/mm) compared to ~ 100 in rIC and very few in the BLA. Considering that glutamatergic PBN neurons project directly to the central nucleus of the amygdala (CeA) rather than the BLA, it is more interesting to examine Fos in CeA rather than (or in addition to) to BLA.

Line 86, no need to capitalize Cholecystokinin

Line 87 Omit "degree of"

Line 131, Why hChR2 rather than ChR2 as used on line 117? What does the "h" signify?

Line 134, I think the abbreviation should be EPMT rather than EMPT. Maybe abbreviation is not needed at all if this test is not used again.

Line 195, "significantly increased body weight" I would recommend saying "prevented further weight loss" instead

Line 216 "All tested cells were evoked with EPSCs, but not with IPSCs." It would be better to say, "Light activation resulted in eEPSCs and no eIPSCs in all cells tested."

Line 224, "which confirms" as "s"

Line 229, Slc17a6 (italics) mRNA rather than Vglut2 (or say mRNA encoding Vglut2)

Line 268 and 270, I would eliminate "visceral" from these sentences

Line 281-285, It is true that lesions of the CeA do not prevent conditioned taste aversion (CTA), perhaps because CTA can be elicited by inputs to either CeA or BNST. Activation of excitatory inputs to the CeA is sufficient to elicit a strong CTA. CCK, LiCl and LPS all activate CGRP neurons in the PBN that project strongly to the CeA, but not necessarily to the PKC delta neurons. Thus, the conclusion of this paragraph is a misleading. The whole paragraph could be eliminated without impacting the main message.

Line 289 Capitalize all 3 words "Ventral Tegmental Area"

Reviewers' comments:

Reviewer #1 (Remarks to the Author):

Unilateral Contribution of the anterior insula to feeding responses to aversive visceral stimuli in adult mice

Revisions R2

Yu Wu, Changwan Chen, Ming Chen, Xinyou Lv, Kai Qian, Haiting Wang, Lifei Jiang, Lina Yu, Min Zhuo, and Shuang Qiu

The manuscript is improved following the reviewers comments. However, we have the below described reservations (mainly for figures 1 and 2).

For all figures:

The Journal suggests that statistical information is included in the Results not the legends.

Response

We have included the statistical information in the Results and deleted them in the Figure Legends.

The time point for c-fos immunohistochemistry and the RNAscope experiments is not stated.

Response

We are sorry for it. We have added the time point for c-fos immunohistochemistry in Methods (p21-22 in the revised manuscript) and that for the RNAscope in Figure Legends (p15 in the Supplementary information).

Figure 1:

1. The authors adjusted the doses for CCK and Ghrelin, as we recommended. From these new experiments (1a), one can note that these agents do induce activation of the insula, yet not in a lateralized fashion like LiCl, LPS and Cisplatin. However, another interesting difference not addressed by the authors is that what the authors term as “physiological agents” (i.e. CCK and Ghrelin), are activating distinct part of the insula than presumably immune response-activating agents (LiCl, LPS, Cisplatin). According to the new results, regardless of lateralization, anorexigenic agents primarily activate deep layers of the Agranular subdivision of the insula, whereas Ghrelin, the only hunger-producing hormone activates almost exclusively the Granular aIC. For these reasons, as we indicated in our first communication, it would be important to define where this change is happening (1a, c).

Response

Thanks for the reviewer's comment. We have systematically analyzed Fos expression in different subdivisions (agranular, dysgranular and granular) of the caudal segment of the right aIC in response to different stimuli and observed that "*LiCl- and LPS-induced Fos⁺ neurons were exclusively located in the agranular part of the caudal segment of the right aIC, whereas CCK-induced Fos⁺ neurons were mainly located in the granular part and Ghrelin-induced Fos⁺ neurons were diffusely distributed in this segment*". We have included these data in the revised manuscript as Supplementary Fig. 2g and h.

2. The authors state that they quantified connectivity and c-fos responses using 4 slices between Bregma 1.18 and 0.00 or 14 slices between Bregma +2.46 and -1.06. This largely over-estimates the homogeneity of the response, while differences between the aIC layers and subregions have been ignored. Complementary studies (Supplementary 4) indicate that the authors may have at least partial answers as to exactly where the lateralization is of relevance.

Response

Thanks for the reviewer's comment. Previously, quantification of Fos staining in the IC was performed on every 6th slice in the IC (thus 4 slices between Bregma +1.18 and 0.00 or 14 slices between Bregma +2.46 and -1.06). To minimize the homogeneity of the response, we performed Fos staining on every third slice in the IC (between Bregma +2.46 and -1.06, therefore totally 28 slices per mice). The result is similar with previous one that the caudal segment of the right aIC is significantly activated by injection (i.p.) of LiCl or LPS (Supplementary Fig. 2a-f). We also systematically analyzed Fos expression in different subdivisions (agranular, dysgranular and granular) of the caudal segment of the right aIC. We have described in details the number of slices we used for staining in the Methods and added these data in the revised manuscript as Supplementary Fig. 2.

3. The images displayed suggest that the CaMKII staining worked partially – perhaps the authors would find it easier to do this using CaMKII-Cre mice at their disposal. The methods indicate that the authors used two rabbit antibodies (c-fos and CaMKII) together, is it a typing error?

Response

We are sorry for this mistake. Fos antibody (SYSY system, 226 003) used in this work is a rabbit antibody, while CamKII antibody (Abcam, ab22609) is a mouse antibody. We have corrected it in the revised manuscript.

To further confirm the cell type in the aIC that responds to aversive stimuli, we firstly injected virus AAV9-CamKII-mCherry into the caudal segment of the right aIC of the wild-type mice. Three weeks later, we observed that 87.8% of Fos⁺ neurons activated by LiCl were colocalized with mCherry positive cells in this site (Supplementary Fig. 3d, e). Next, we injected (i.p.) LiCl into the *CamKII-Cre::Ai14 (tdTomato)* mice and performed immunostaining 1.5 hr after injection. We observed that 92.5% of Fos⁺ neurons induced by LiCl injection were also tdTomato positive

neurons, confirming that most of the activated neurons are excitatory neurons expressing CamKII (Figure 1 in this rebuttal letter).

Figure 1. LiCl-induced Fos expression in the aIC of *Camk2a-Cre::Ai14* mice. (a) Representative histology of Fos-like immunoreactivity in the left and the right aIC of *Camk2a-Cre::Ai14* mice. Scale bar, 100 μ m. (b) Quantification of Fos⁺ neurons and the colocalization of Fos⁺ neurons (green) with CamKII⁺ neurons (red)(n = 4 mice per group)

Figure 2:

1. The construct that the authors claimed to have injected to chemogenetically inhibit the projection cannot be a ChR2 expressing AAV (AAV8-DIO-hChR2(H134R)-eGFP), as mentioned in the results section?

Response

We are sorry for this typing error. It should be “optogenetically”, not “chemogenetically”. We have corrected it in the revised manuscript.

2. Optogenetic activation of the projection does seem to suppress feeding responses quite convincingly (f). However, the control experiments in the optogenetic inhibition experiment might require better matching with the stimulated condition. The trials shown indicate that control light exposure, eventually leads to feeding, so it is perhaps best to use a different representative trial that reflects the clarity of the differences in the data analyzed.

Response

We appreciate the reviewer for pointing it out and giving us an opportunity to clarify it. The mouse used here (Fig. 2f) was 24-hr fasted and had a tendency to eat more during the test. Moreover, control light was triggered after feeding initiation. So, the control trials shown look like control light leads to feeding. To avoid obscurity, we used another representative trial in the revised manuscript.

3. There is no information regarding the number of approaches or the percentage spent feeding. It is further unclear whether mice used here were fasted or fed and what the hypotheses entailed in these experiments were.

Response

We are sorry for missing the information. We have modified the related part, included the information about the number of approaches, the percentage spent feeding and the status of the mice used for different tests, and described the details in the Results (p7-8) and Methods (p24) part in the revised manuscript.

4. The data regarding fibre optometry in Figure 2 very elegantly demonstrate the unilaterality of the response to LiCl and cisplatin. However, what is also apparent is that LiCl or cisplatin stimulation increases activity in the projection at distinct time points (10seconds vs 30minutes). Given this, it is further highlighted that control stimulation should be more closely matched to the timing of the actual stimulation be useful.

Response

Thanks to the reviewer's comment. We have observed that *in vivo* photostimulation instantly activated the aIC neurons within millisecond-timescale (Fig. 2b), which then rapidly induced the change of feeding behavior (Fig. 2d and 2f). In contrast, LiCl and Cisplatin stimulation (i.p) unilaterally increased insular activity at distinct time points (10 seconds vs 30 minutes, Fig. 1), which may due to the different characteristics of these two reagents, such as the rate of absorption and diffusion, the way of action, the circuit involved, etc. Combined together, we agree with the reviewer that the timing of the stimulation (control *v.s* actual) should be closely matched. Additionally, we believe that timing is a critical factor that should be taken into consideration especially when we plan to use optogenetic manipulation of the neuronal activity to mimic the physiological or pathological stimulation.

Figure 3-6: OK.

Discussion:

1. We do not agree with the over-interpretation regarding unphysiological and physiological stimuli. Given the evidence provided by the authors themselves, the insula shows clear responses to both ghrelin and CCK (compared to saline), both physiological stimuli. Previous studies in the cytoarchitecture or the insula in fact suggest that visceral information should primarily activate the granular insula, which can be seen in the ghrelin responses. Conversely, a number of other studies suggest the dysgranular/agranular subregions to be dedicated to emotional processing, as well as CS-US associations.

Response

Thanks to the reviewer's comment. We agree with the reviewer that the aIC shows response not

only to LiCl, but also to ghrelin and CCK and it is inappropriate to confine it to pathological stimulation. We have modified the related description and deleted the description related to physiological or pathological stimuli in the revised manuscript.

2. The aspect of local connectivity at the aIC was not addressed sufficiently, while calling the circuit a “visceral hot spot” might be problematic since the study did not use a method to cover the IC continuously.

Response

Thanks to the reviewer’s comments. Although we have used more slices for staining to minimize the homogeneity, we agree with the reviewer that “visceral hot spot” may be problematic. We have deleted it or replaced it with “segment” in the revised manuscript.

Reviewer #2 (Remarks to the Author):

The authors had sufficiently addressed my concerns over the previous version of this manuscript. I therefore support the publication of this study.

Response

We sincerely appreciate the reviewer for their positive comments.

Reviewer #3 (Remarks to the Author):

The authors have attended to most of my concerns. I have provided some line-by-line editorial suggestions below. The only new information requested deals with Fos induction in CeA in response to LiCl, LPS and Cis-platin. The authors probably have that information since the quantified Fos in the adjacent BLA.

Response

Thanks for the reviewer’s suggestion. We have included these data (Fos expression in CeA in response to LiCl, LPS and Cisplatin) in Supplementary Fig. 1.

Line 44, Consider revising your sentence: “By now, it is comparatively elusive as to the neural circuits involved in mediating feeding responses under these non-homeostatic conditions” to the following: “However, the neural circuits involved in mediating feeding responses under these non-homeostatic conditions are comparatively elusive”

Response

Thanks for the suggestion. We have revised this sentence accordingly.

Line 80, The authors show robust activation of Fos in the PBN (~ 400 Fos+ cell/mm) compared to ~ 100 in rIC and very few in the BLA. Considering that glutamatergic PBN neurons project directly to the central nucleus of the amygdala (CeA) rather than the BLA, it is more interesting to examine Fos in CeA rather than (or in addition to) to BLA.

Response

Thanks for this valuable suggestion. We have detected Fos expression in the CeA and observed robust activation of Fos in the CeA in response to LiCl stimulation (Supplementary Fig. 1).

Line 86, no need to capitalize Cholecystokinin

Response

We have corrected it.

Line 87 Omit “degree of”

Response

We have corrected it.

Line 131, Why hChR2 rather than ChR2 as used on line 117? What does the “h” signify?

Response

Here, “h” (hChR2 or humanized ChR2) signifies the replacement of the algal codons with mammalian codons in order to achieve higher expression levels^{1,2}.

Line 134, I think the abbreviation should be EPMT rather than EMPT. Maybe abbreviation is not needed at all if this test is not used again.

Response

We agree with the reviewer that this abbreviation is not needed since this test is not used again. We deleted it in the revised manuscript.

Line 195, “significantly increased body weight’ I would recommend saying “prevented further weight loss” instead

Response

Thanks to the comments. We have modified this sentence as suggested.

Line 216 “All tested cells were evoked with EPSCs, but not with IPSCs.” It would be better to

say, ***“Light activation resulted in eEPSCs and no eIPSCs in all cells tested.”***

Response

Thanks to the comments. We have modified this sentence as suggested.

Line 224, “which confirms” as “s”

Response

Thanks to the comments. We have corrected it.

Line 229, Slc17a6 (italics) mRNA rather than Vglut2 (or say mRNA encoding Vglut2)

Response

Thanks to the comments. We have corrected it.

Line 268 and 270, I would eliminate “visceral” from these sentences

Response

Thanks to the comments. We have deleted it.

Line 281-285, It is true that lesions of the CeA do not prevent conditioned taste aversion (CTA), perhaps because CTA can be elicited by inputs to either CeA or BNST. Activation of excitatory inputs to the CeA is sufficient to elicit a strong CTA. CCK, LiCl and LPS all activate CGRP neurons in the PBN that project strongly to the CeA, but not necessarily to the PKC delta neurons. Thus, the conclusion of this paragraph is a misleading. The whole paragraph could be eliminated without impacting the main message.

Response

We agree with the reviewer’s comment and have deleted this paragraph in the revised manuscript.

Line 289 Capitalize all 3 words “Ventral Tegmental Area”

Response

Thanks to the comments. We have corrected it.

Reference

1. Yizhar O, Fenno LE, Davidson TJ, Mogri M, Deisseroth K. Optogenetics in neural

systems. *Neuron* **71**, 9-34 (2011).

2. Zhang F, Wang LP, Boyden ES, Deisseroth K. Channelrhodopsin-2 and optical control of excitable cells. *Nat Methods* **3**, 785-792 (2006).

REVIEWERS' COMMENTS:

Reviewer #1 (Remarks to the Author):

I have no further comments, the manuscript is improved and ready for publication